| Open Peer Review | Environmental Microbiology | Methods and Protocols
# Using phenotyping to visualize and identify selfish bacteria: a methods guide

G. Reintjes,[1,2] G. Giljan,[1] B. M. Fuchs,[1] C. Arnosti,[3] R. Amann[1]

**ABSTRACT** Polysaccharides are dominant components of plant and algal biomass, whose degradation is typically mediated by heterotrophic bacteria. These bacteria use extracellular enzymes to hydrolyze polysaccharides to oligosaccharides that are then also available to other bacteria. Recently, a new mechanism of polysaccharide processing—"selfish" uptake—has been recognized, initially among gut-derived bacteria. In "selfish" uptake, polysaccharides are bound at the outer membrane, partially hydrolyzed, and transported into the periplasmic space without loss of hydrolysis products, thus limiting the availability of smaller sugars to the surrounding environment. Selfish uptake is widespread in environments ranging from the ocean's cool, oxygen-rich, organic carbon-poor waters to the warm, carbon-rich, anoxic environment of the human gut. In this methods paper, we present a detailed guide to identifying selfish bacteria, including techniques for rapidly visualizing selfish uptake in complex bacterial communities, detecting selfish organisms, and distinguishing their activity from that of other community members.

**IMPORTANCE** Understanding the role of heterotrophic bacteria in the degradation of organic matter is critical for comprehending carbon cycling and microbial ecology across different environments. This study highlights the significant prevalence of "selfish uptake" among bacteria—often overlooked by standard microbial activity assessments—and presents the method used to quantify and identify these "selfish" bacteria. Found in diverse habitats such as anoxic gut environments, oxygenated waters, sediments, and soils, their widespread presence underscores the necessity of revisiting current methodologies to include these crucial organisms. By identifying and studying selfish bacteria, we can gain detailed insights into how microbial communities function, how carbon flows through ecosystems, and how these processes impact global biogeochemical cycles.

**KEYWORDS** selfish uptake, carbon cycling, enzymatic hydrolysis, bacterial community function, flow sorting, polysaccharide degradation

Polysaccharides constitute the largest pool of metabolically accessible organic carbon in the biosphere (1). Their primary sources are phototrophic organisms of terrestrial and marine environments, such as plants and algae, which produce polysaccharides as structural complexes and as storage compounds (2–4). Polysaccharides account for about half of the living biomass of phytoplankton (3) and terrestrial plants (5) and represent a major fraction of the immense reservoir of detrital organic matter in soils (6), sediments (7), and seawater (8). The cycling of polysaccharide-derived material thus is critical for processes and issues ranging from the global flux of carbon to human (9–11) and animal (12, 13) nutrition.

Polysaccharide degradation, transformation, and remineralization are mainly performed by bacteria, which are abundant in the environment (14) and in the digestive

**Peer Reviewer** Mirjam Czjzek, Biological Station Roscoff, Roscoff, France

Address correspondence to G. Reintjes, reintjes@uni-bremen.de.

G. Reintjes and G. Giljan contributed equally to this article. Author order was determined in order of decreasing seniority.

The authors declare no conflict of interest.

See the funding table on p. 16.

tracts of animals (15). In terrestrial ecosystems, fungi also contribute substantially to the degradation of plant polysaccharides, such as cellulose (16, 17). Degradation of polysaccharides is challenging for bacteria because some polysaccharides are structurally complex (18), containing different monosaccharides connected by a wide range of glycosidic linkages (5, 18). Since these monosaccharides can be linked together via any of five or six positions, the structural complexity of polysaccharides far outpaces that of other biopolymers, such as proteins. Thus, correspondingly complex enzymatic systems are required for polysaccharide degradation (19).

Initial enzymatic hydrolysis of polysaccharides by bacteria occurs outside the cell due to the large size of polysaccharides. This extracellular hydrolytic step produces lower molecular weight products that can be released into the surrounding environment and may be available for uptake by organisms that did not produce the extracellular enzymes (Fig. 1A, C, and E) (20–22). This potential loss of hydrolysis products constitutes a complication for extracellular enzyme-producing bacteria, which need to obtain sufficient hydrolysate as a return on their investment in hydrolytic enzymes. Recently, however, a distinctly different mechanism of polysaccharide processing—"selfish" uptake—has been recognized (Fig. 1A, B, and D), initially in gut bacteria. "Selfish" bacteria (23) bind polysaccharides and partially hydrolyze them to oligosaccharides, which are transported into the periplasm and then undergo further degradation. This minimizes the release of mono-, di-, and tri-saccharides into the surrounding environment, ensuring a return on their enzymatic investment.

The cost of enzyme production and the complexity of enzymatic systems required to deconstruct many polysaccharides, therefore, may be balanced in different ways. Selfish uptake likely requires high energetic investment to express many enzymes but is characterized by little loss of hydrolysis products (23). External hydrolysis potentially leads to the loss of low molecular weight hydrolysis products to other organisms but might be coordinated among bacteria (e.g., via quorum sensing [25], such that enzyme production and hydrolysate uptake can be optimized within a community). Initial assessments of the prevalence of selfish uptake and external hydrolysis in the ocean suggest that strategies of substrate processing change with location, as well as with the nature and abundance of substrates (22, 26, 27). In particular, selfish uptake may pay off, particularly in cases where competition for a specific substrate is very high, as well as in cases where the abundance of a complex substrate is low, such that a return on investment in complex enzymatic systems needs to be guaranteed (28).

In sum, "selfish uptake" is prevalent among organisms found in the anoxic, organic-carbon-rich gut environment (29), and also in the oxygenated organic carbon-poor waters of the surface ocean (30). The recent discovery that selfish bacteria are also abundant throughout the oceanic water column and take up substrates that are not hydrolyzed externally demonstrates that standard methods to determine microbial activities may overlook important processes and organisms (31). In short, given their presence in these distinctly different environments, selfish bacteria are likely to be found in many other natural environments, including sediments, soils, and digestive tracts of a wider range of organisms. Therefore, detecting the presence and activities of selfish bacteria is central to our efforts to understand carbon cycling, animal nutrition, and the microbial ecology of a wide range of environments. Fortunately, detecting the presence of selfish bacteria and selfish activity experimentally is a straightforward process.

We emphasize here that selfish bacteria transport large polysaccharide fragments into the periplasmic space; they do not simply bind them to the outer membrane. Super-resolution light microscopy has shown that FLAPS staining is confined to the periplasmic space, which can be well-defined by the simultaneous use of a membrane stain (30). Fluorescence line profiling and z-stack images localizing the 3 dimensions of the cell demonstrate that the polysaccharide is within the outer membrane but outside the cell wall (30). In addition to visual evidence, physiochemical proof of polysaccharide uptake is provided by work with mutant strains lacking the outer membrane uptake system for

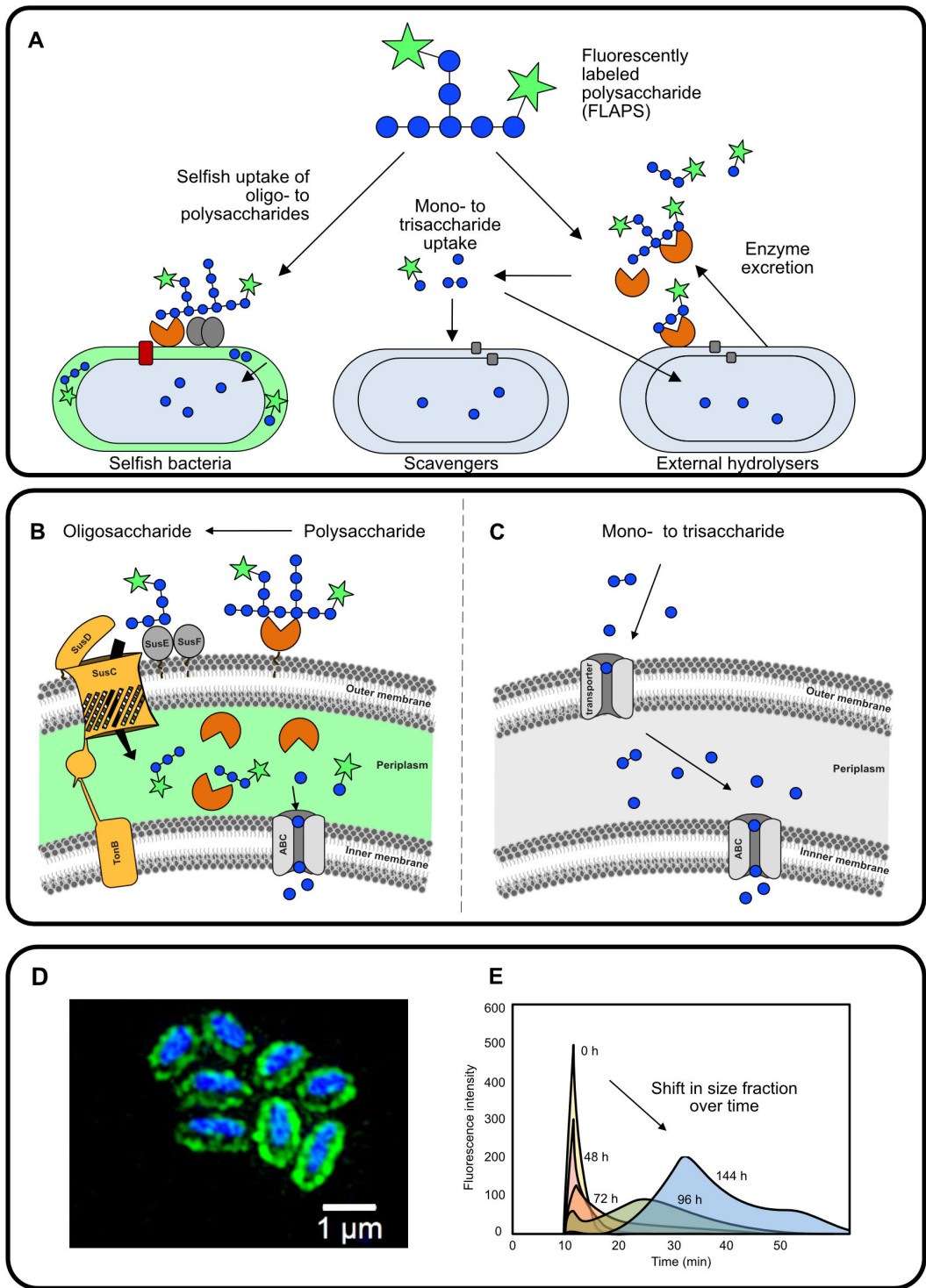

**FIG 1** Heterotrophic utilization of polysaccharides as shown with a fluorescently labeled polysaccharide (FLAPS). (A) Schematic overview of the two known polysaccharide utilization mechanisms—selfish uptake and extracellular hydrolysis with subsequent uptake of monosaccharides and small oligosaccharides. (B) Conceptual schematic of selfish FLAPS uptake into the periplasm of a cell where it is further hydrolyzed to monosaccharides that are transported through the inner membrane into the cell. (Adapted from Hehemann *et al.,* (13) (C) Conceptual schematic of enzymatic extracellular FLAPS hydrolysis into monosaccharides that are transported into the cell. (Adapted from Arnosti *et al.,* (21) (D) Microscopic visualization of selfish FLAPS uptake and accumulation in the cells. The cell DNA is shown by DAPI staining in blue; the FLAPS is shown in green. Scale bar = 1 µm. (E) Gel permeation chromatograms showing systematic changes in molecular weight of FLAPS with incubation time (0–144 h). This chromatogram shows the method used to measure the external (outside of the cell) hydrolysis of FLAPS. (Adapted from Arnosti, 2003 (24).

polysaccharides (SusC/D) or the full PULs. Unlike their wild-type counterparts, these mutant strains were not stained with FLAPS (13, 29).

The mechanisms of selfish uptake of polysaccharides have been thoroughly studied among members of the *Bacteroidota*, especially gut-associated strains. Studies have demonstrated that polysaccharides are bound at the outer membrane, partly hydrolyzed, and then transferred into the periplasmic space, where they are further hydrolyzed (Fig. 1B) (23, 32–34). Genes involved in selfish uptake by *Bacteroidota* are typically possessed in PULS (21, 32), including the SusC/D transport system. Observations that selfish uptake is also carried out by organisms lacking PULs (e.g., *Planctomycetes*) and organisms lacking SusD (*Gammaproteobacteria*) demonstrate, however, that the presence of a PULs or a SusC/D system is not a strict requirement for selfish uptake. Moreover, bacteria with PULs and SusC/D systems may carry out external hydrolysis in addition to selfish uptake (33, 34). Examination of genomes without experimental incubations consequently is not sufficient to demonstrate selfish behavior. The specific means by which organisms lacking the SusC/D system carry out selfish uptake remain to be determined. Additionally, the current concept of selfish uptake requires a cell organization with an outer membrane and is therefore restricted to gram-negative bacteria. In sum, the broad use of a selfish strategy of substrate processing in the environment—partly by organisms whose specific mechanism of selfish uptake is not yet known—suggests that much is waiting to be revealed.

Identifying the presence of selfish bacteria, moreover, opens the door to further focused investigations, starting with the taxonomic identification of selfish bacteria and extending to flow cytometric methods that enable the physical separation of these bacteria and further analysis of their physiology, biochemistry, and activity. We present an example from the North Sea demonstrating how hunting for selfish bacteria can yield further information about community activities, identities, and carbon flow in a natural system. These data, specifically the analysis of selfish uptake, FISH and extracellular hydrolysis rates, were initially presented in Giljan *et al*. (35); here, we present in detail results that were not discussed at length in that manuscript. We also discuss additional insights from human gut microorganisms (13). The approach we used could easily be applied to studies in fields ranging from animal nutrition to terrestrial and aquatic investigations of the ecology of microbial communities and the pathways of carbon degradation that they catalyze in natural environments. Overlooking selfish bacteria and their activities in any environment means that we are overlooking important organisms, as well as pathways of material flow and energy transfer. Here, we present in detail the methods required to reveal their presence.

## MATERIALS AND METHODS

### Sample collection and substrate incubation

Surface seawater was collected at the long-term ecological research station Helgoland Roads on the 17th and 19th September 2018. On both dates, FLAPS incubation experiments were conducted in sterile, acid-rinsed glass bottles in the dark at 18°C (ambient water temperature). Incubations with FLA-laminarin, FLA-xylan, and FLA-chondroitin sulfate at a final concentration of 3.5 µM monomer equivalent were conducted in triplicates. Incubations without FLAPS addition and a single autoclaved (killed) control per substrate were also included. After 0, 1, 6, 12, 24, 48, and 72 h of incubation, subsamples were taken for total cell counts, analysis of selfish uptake and external polysaccharide hydrolysis, FISH analysis, and fluorescence-activated cell sorting. Subsamples for bacterial community analysis were taken at 0, 24, and 72 h. Quantitative data on phytoplankton community composition (cell counts of centric and pennate diatoms, dinoflagellates, coccolithophores, and flagellates), and data on water temperature and salinity, nutrient availability (silicate, nitrate, and phosphate), and Chl *a* concentrations were collected on the same dates (35).

## Synthesis and characterization of fluorescently labeled polysaccharides (FLAPS)

Many polysaccharides can be used for FLAPS experiments (Table S1). The overall procedure is described in great detail in (24). In brief, FLAPS are prepared using purified polysaccharides obtained from commercial sources or extracts of phytoplankton, for example, purified to a specific molecular weight cutoff. The polysaccharides are dissolved in water and then activated using a cyanogen bromide (CNBr) solution while monitoring the pH of the reaction mixture. Small volumes of NaOH solution are added to maintain the pH of the reaction mixture above 9.0 for 5 min. The CNBr reacts with hydroxyl groups of the polysaccharide to form cyanate esters, which will react readily with primary amines. To separate the activated polysaccharide from the residual CNBr and stabilize the polysaccharide, after 5 min, the reaction volume is injected into a gel permeation chromatography column running with a borate buffer. The activated polysaccharide is collected from the column, and then incubated for 12–24 h at room temperature with the fluorophore fluoresceinamine (Fluoresceinamine, Isomer II, Sigma-Aldrich). During labeling, the primary amine group on the fluoresceinamine linker forms a stable isourea linkage with the activated polysaccharide. We note that although other primary amine-containing fluorophores can in principle be used to label polysaccharides, fluorophores with an ester group in their linker arm are unsuitable for experiments in aqueous solution because the ester group is very easily auto-hydrolyzed in solution (personal communication). After activation, the labeled polysaccharide is separated from the unreacted fluorescent tag using size exclusion chromatography or centrifugation with membrane cartridges (24, 36, 37).

In principle, any soluble polysaccharide (or phytoplankton extract or phytoplankton-derived DOC [24, 38]) can be labeled using this procedure. The primary limitations on polysaccharide choice are the requirements for solubility in an aqueous solution and stability (and solubility) at pH >9.5, since pH varies slightly during the synthesis procedure. The labeling density of FLAPS is highly polysaccharide dependent (likely related to the hydrodynamic volume and conformation of a polysaccharide and thus to access of a fluorophore to activated polysaccharide sites). It must be determined for each batch of FLAPS: carbohydrate content can be measured using the phenol-sulfuric acid method, and fluoresceinamine content can be measured via absorbance at 430 nm. Labeling with a fluorescent tag is not thought to change the bioactivity of a polymeric compound (37). Note that other labeling methods that specifically target the anomeric carbon (the end of a polysaccharide chain with a free anomeric carbon) have to date not been usable for detection of selfish uptake (personal communication). We hypothesize that the binding proteins that are part of the selfish uptake system are unable to accommodate the terminal end of a polysaccharide that bears an attached fluorophore.

## Measurements of external (extracellular) hydrolysis

The rate of extracellular FLAPS hydrolysis is measured by the change in size distribution from the initial polysaccharide to smaller hydrolysis products, determined via gel permeation chromatography, as shown in Fig. 1E and described in detail in Arnosti (2003) (24).

External hydrolysis of FLAPS was measured by analyzing filtrate from the samples collected for bacterial community analysis (described below). These samples, collected after 0, 24, 48, and 72 h of incubation, were analyzed as described in detail in Arnosti (2003) (24). The earlier time points (6 and 12 h) were not measured because, in our experience, environmental samples from the water column do not show sufficient activity for hydrolysis to be detected at these early time points. Sediment incubations and incubations with pure cultures of bacteria, in contrast, typically show much higher hydrolysis rates, and hydrolysis can be measured on timescales of hours (24).

## Selfish uptake measurements

For all microscopic analysis, samples were fixed with formaldehyde at a final concentration of 1%, filtered onto a 0.2 µm pore size polycarbonate filter, counterstained with 4′,6-diamidino-2-phenylindole (DAPI), and mounted in a Citifluor/VectaShield (4:1) solution.

Total DAPI counts and selfish polysaccharide uptake were analyzed using a fully automated epifluorescence microscope (Zeiss AxioImager.Z2, Carl Zeiss), equipped with a with a cooled charged-coupled-device (CCD) camera (AxioCam MRm; Carl Zeiss) and a Colibri LED light source (Carl Zeiss) with three light-emitting diodes (UV-emitting LED, 365 ± 4.5 nm for DAPI; blue-emitting LED, 470 ± 14 nm for FLAPS 488; red-emitting LED, 590 ± 17.5 nm for the tyramide Alexa 594, FISH), combined with the HE-62 multifilter module (Carl Zeiss). This module consists of a triple emission filter TBP 425 (± 25), 527 (± 27), LP 615, including a triple beam splitter of TFT 395/495/610. Images were acquired using a 63× oil immersion plan apochromatic objective with a numerical aperture of 1.4 (Zeiss) at the selected wavelengths specified above. Images from the substrate channel (FLAPS, 470 nm) were acquired at three exposure times (10 ms, 35 ms, and 140 ms) to cover the diversity of signal intensities and patterns of FLAPS accumulation. A minimum of 45 fields of view were acquired, and microscopic images were exported into the modified image analysis software ACMEtool (M. Zeder, Technology GmbH, http://www.technobiology.ch and Max Planck Institute for Marine Microbiology, Bremen), and signals were evaluated (Table S2) according to Bennke et al. (2016) (39). Briefly, a positive signal in DAPI and FLAPS images with a minimum overlap of 30%, as well as a minimum signal background ratio of 1, were required for positive identification as a FLAPS-stained cell. An overlap of all three signals indicated a FLAPS-stained cell identified by a specific FISH probe (see below). Each image was also recorded on the Auto signal (590 nm) with an exposure time of 300 ms to visualize potential autofluorescence from pigments within cells. All bacteria that had a positive signal on the Auto, DAPI, and FLAPS channels were recognized as Cyanobacteria and excluded from the calculations for selfish uptake.

For single-cell detection of substrate uptake patterns, cell membranes stained with Nile Red and FLA-substrate labeled cells were subsequently visualized using SR-SIM. Individual cells were analyzed with a Zeiss ELYRA PS.1 (Carl Zeiss) microscope equipped with 561, 488, and 405 nm lasers and BP 573-613, BP 502-538, and BP 420-480 + LP 750 optical filters. A Plan-Apochromat 63×/1.4 Oil objective was used to take z-stack SR-SIM images with a CCD camera. Data processing and image analysis were done using the ZEN software package (Carl Zeiss).

## FLAPS background signals: differentiating between signal and noise

Although the application of the FLAPS substrate incubations is straightforward, differentiating between a substrate-stained cell and substrate background noise requires practice. The accumulation of FLAPS in the cells' periplasm results in a regular outline of the cells' morphology; therefore, irregular shapes hint at the presence of background noise. Nevertheless, different patterns in FLAPS accumulation within the cells lead to variations in the distribution and intensity of the FLAPS signal (31). Total cell counts should be determined for each incubation time point to test if a drop in selfish activity correlates with an increase in cell number, since the distribution of substrate to daughter cells might lead to fluorescence dilution within a cell after division.

## Background signal test of the FLAPS: investigating non-specific staining

The application of FLAPS in seawater or salt-containing medium can result in the production of non-cell associated background signals (Fig. S5) through FLAPS adhesion to *in situ* particulate organic matter or transparent exopolymeric substances. These unspecific signals must be differentiated from true signals to achieve accurate quantification of FLAPS uptake. An essential identification point of FLAPS-stained cells is the co-localization of a DAPI (cell DNA) signal with a FLAPS signal. This can help differentiate unspecific binding and particles from the cell.

Three polysaccharides that frequently show unspecific substrate signals (xylan, laminarin, and pullulan) were used to test background signal reduction. We tested several alterations to the standard FLAPS protocol (Table S4). All FLAPS were mixed with 10 mL (A) seawater from sampling station Kabeltonne off the island Helgoland, (B) 1× sterile-filtered artificial seawater (ASW), and (C) 18 MΩcm water (MQ-water) at a concentration of 3.5 µM monomer equivalent final concentration. After inoculation, all solutions were supplemented with 1% Formaldehyde (FA) and fixed at room temperature for 1 h. An unfixed non-reference was prepared for (D) ASW and (E) MQ-water. To check if the direct contact of the highly concentrated polymers with bivalent cations catalyzes a polymerization process of the substrates, the FLAPS were diluted 100-fold with MQ before the addition to (F) Helgoland water. For FLA-xylan, FLA-laminarin, and FLA-pullulan, ethylenediaminetetraacetic acid (EDTA) as an anti-chelating agent was added at a final concentration of 25 nM to the seawater (G), ASW (H), and MQ-water (I) to compete for the $Ca^{2+}$Ions and potentially reduce coagulation. To avoid the addition of already coagulated substrates, FLAPS solutions were warmed to 32°C for 5 or 15 min before addition to ASW (J, K), MQ (L, M), or seawater (N, O), respectively (Table S4; Note: heating FLAPS is not recommended, see below). Each 10 mL sample was filtered onto a 0.2 µm pore size polycarbonate filter, and microscopic pictures were taken for automated image analysis as described in detail in the methods section.

The substrate diluted in seawater with subsequent FA fixation (Fig. S5 left panel; Table S4A) was taken as the reference for a standard incubation in the marine environment. When laminarin, xylan, and pullulan were diluted in sterile artificial seawater, all background signal from unspecific binding to organic matter from an environmental sample was removed for pullulan but not for laminarin and xylan. To test whether bivalent cations from the seawater initialize the coagulation process, the FLAPS were added to sterile filtered MQ water. We found that the dilution of FLAPS in ultrapure water was a major cause of increased background signals if formaldehyde was added in addition, but the addition of formaldehyde did not appear to cause additional background signals in seawater. Neither the filtration of the substrate stock to remove particles larger than 0.22 µm from the substrate stock before the addition to the incubation nor the addition of EDTA as an anti-chelating agent led to a decrease in substrate background signal. EDTA addition led to the even distribution of the FLA substrate across the whole filter and led to complete overexposure (marked with N.D. in Table S4).

Additionally, we tested the pre-warming of the substrate stock to dissolve potential aggregates of polysaccharides. Pre-warming of the substrate stock for 5 or 15 min to 32°C did not remove the background signal. The background signal slightly decreased in number after 15 minutes at 32°C for laminarin and xylan. Note, however, that not all FLAPS are heat stable; we do not recommend routinely heating FLAPS substrates. If they are used in temperature experiments (e.g., 40, 41), FLAPS should be tested prior to use to establish the range of temperature stability of the polysaccharides.

## Flow cytometry

Flow cytometry can also be used to detect selfish uptake. However, as the detection time is in the microsecond range (42), sensitivity is lower than in a good microscope, and therefore, strong staining of the target cell compared with background signals is necessary.

Single-cell fluorescence quantification was determined using an Accuri C6 flow cytometer (BD Accuri Cytometers, USA). The 8- and 6-peak validation beads (Spherotech, USA) were used for reference. All culture samples were measured under 488 nm laser excitation, and the fluorescence was collected in the FL1 channel (530 ± 30 nm). The medium with and without fluorescent substrate and an electric threshold of 17,000 FSC-H was used to set the background noise. All bacterial samples, with and without FLAPS, were measured using a slow flow rate with a total of 10,000 events per sample in triplicate. Bacteria are detected from the signature plot of SSC-H vs. green fluorescence

(FL1-H). The flow cytometric output was analyzed using FlowJo v10-4-2 software (Tree Star, USA).

For background signal visualization in a flow cytometer, 2 mL of 1% FA fixed seawater from the Helgoland autumn sampling was each mixed with the fluorescently labeled laminarin, xylan, and pullulan, at a final concentration of 3.5 µM monomer equivalent and analyzed in a BD Influx™ Cell Sorter (Fig. S3C through I). The substrate-specific background signal can be seen in comparison to an unamended treatment control (Fig. S3A) and used as a negative control to separate it from FLAPS-stained cells. However, the comparison of the same sample, incubated with FLA-laminarin for 24 h, showed that indeed the microbial community changes the substrate signature over the time of the incubation (Fig. S3B and C).

## Bacterial community analysis and community statistics

In seawater incubation experiments, the initial bacterial community and changes in community composition and abundance were determined by 16S rRNA amplicon sequencing. At each time point, a subsample of 10 mL from each incubation bottle was filtered through a 0.2 µm pore size polycarbonate filter; the filtrate was used to analyze extracellular hydrolysis rates, as described above. Total DNA was extracted from the filter with the DNeasy Power Water Kit (Quiagen), and the hypervariable V3-V4 region (490 bp) of the 16S rRNA was amplified from the DNA using the S-D-Bact-0341-b-S-17 and S-D-Bact-0785-a-A-21 (43) primer pair with an Ion Torrent sequencing adapter and an Ion Xpress Barcode Adapter (Thermo-Fischer Scientific) attached to the forward primer. The PCR product was purified, and the remaining free primers were removed using the AMPure XP PCR Cleanup system (Beckman Coulter). A pool of barcoded PCR products in equimolar concentration was further amplified in an emulsion PCR with the Ion Torrent One-Touch System (Thermo FiFishercientific). Sequencing was done on an I Ion TorrentGM sequencer (Thermo FiscFisherentific) in combination with the High-Q™ View chemistry (Thermo Fischer Scientific). Quality trimmed (> 300 bp sequence length, <2% homopolymers, <2% ambiguities) reads were demultiplexed and used as input for the SILVAngs pipeline (44) for taxonomic assignment of the reads based on sequence comparison to the SSU rRNA SILVA database 312.

## Fluorescence *in situ* hybridization for taxonomic identification

Combining FISH with FLAPS incubations allows the correlation of taxonomy and function, as defined by the capability of bacteria for selfish polysaccharide uptake.

We tested the effect of two FISH procedures on FLAPS signals. For this, we took two seawater samples from the sampling station Kabeltonne off the island of Helgoland and incubated them for 48 h with (i) FLA-laminarin and (ii) FLA-xylan. Subsequently, we fixed the samples with formaldehyde at a final concentration of 1% for 1 h. We applied a tetra-labeled FISH (45) and the CARD-FISH (46) protocol to the incubations to test if FISH influences the microscopic evaluation of FLAPS-stained cells (see Methods for details). For both procedures, we used probes for the taxonomic identification of most Bacteria, *Planctomycetes*, and *Verrucomicrobiota* (EUB388-I, PLA46, and EUB388-III, respectively, Table S3). Formamide concentrations in the hybridization buffer were probe-specific (Table S3). The number of FLAPS-stained cells after FISH treatment, the co-localization of the FLAPS and FISH signal, and the taxonomic correlation of FLAPS-labeled cells were evaluated using epifluorescence microscopy combined with automated image analysis. As described below, both FISH procedures caused wash-out of FLAPS signal from the cells, which was dependent on the harshness and number of steps in the procedure (Fig. S2 and S4). We recommend the use of the tetra-labeled FISH procedure to minimize signal loss.

Based on the results of the methodological comparison, the abundance of FLAPS-stained *Gammaproteobacteria*, *Bacteroidota*, *Verrucomicrobiota,* and *Planctomycetes* was analyzed on samples from Helgoland FLA-laminarin and FLA-xylan incubations using 4 x

Atto594 labeled probes (GAM42a, CF319a, EUB388-III + competitor EUB338-II, and PLA46, respectively, Table S3).

## A detailed comparison of identifying FLAPS-stained cells with tetra-labeled FISH and CARD-FISH: tetra-labeled FISH is the better choice

We systematically compared the effect of performing tetra-labeled FISH and CARD-FISH on FLAPS-stained cells. Tetra-labeled FISH (referred to in the following as FISH) is fast, consisting of one hybridization and two washing steps. However, the four fluorophores per probe restrict the signal intensity, and therefore, the probe concentration must be high to saturate all possible binding sites ($0.84$ µMol ~ 5 ng DNA µL)$.^{-1}$). Comparatively, the CARD-FISH protocol includes more steps: embedding, cell permeabilization, inactivation, and a catalyzed reporter deposition (CARD)-reaction step. The CARD reaction activates numerous fluorochromes per probe—enhancing signal intensity compared to directly labeled probes—at a lower probe concentration ($0.028$ µM or $0.16$ ng DNA µL$^{-1}$). Ethanol washing steps must be left out of both protocols as ethanol's permeabilization of the cell wall was found to lead to FLAPS signal loss.

For these experiments, we took two seawater samples from the sampling station Kabeltonne off the island of Helgoland, Germany. We incubated them for 48 h with FLA-laminarin and FLA-xylan. Subsequently, we fixed the samples with formaldehyde at a final concentration of 1% for 1 h at room temperature. The cells were then filtered onto 0,2 µm pore size polycarbonate filters using a gentle vacuum of <200 mbar. We applied FISH and CARD-FISH with the probes for the taxonomic identification of most Bacteria, *Planctomycetes*, and *Verrucomicrobiota* (EUB388-1, PLA46, and EUBI388-III, respectively, Table S3). Formamide concentrations in the hybridization buffers were probe-specific (Table S3). After the FISH treatments, all samples were counterstained with 4′,6-diamino-2-phenylindole (DAPI) and mounted in a Citifluor/VectaShield (4:1) solution. We co-localized the DAPI (DNA) FLAPS and FISH signal and evaluated the cellular abundance using epifluorescence microscopy with automated image acquisition and enumeration software (39).

First, we found that the FISH and CARD-FISH treatments caused 4% ± 3% and 13% ± 4% loss of the total cell signals (DAPI), respectively (Fig. S4A1 and 2). Furthermore, FISH caused a loss of 48% ± 3% and 52% ± 3% of the FLAPS signal (laminarin- and xylan-positive signals, respectively, Fig. S4B). Comparatively, a minimum of 71% ± 2% FLAPS signals were lost after the CARD-FISH protocol. (Fig. S4B1 and 2).

Due to the high FLAPS signal loss during CARD-FISH, we recommend using a tetra-labeled probe with the FISH protocol after Manz *et al.* (1992) (45) to identify FLAPS-stained cells. It should be noted that even with FISH, there is a FLAPS signal loss and that the numbers of FISH- and FLAPS-positive cells are likely underestimated (Fig. S3). Optimizations of the FISH protocol to reduce signal loss should be performed. Furthermore, FISH staining (594 nm) can cause crosstalk that facilitates false-positive detection of FISH signals as substrate signals (488 nm). We recommend an emission filter with a reduced bandwidth for the green spectrum.

## RESULTS AND DISCUSSION

### A rapid workflow to detect active polysaccharide utilizers

Fluorescently labeled polysaccharide (FLAPS) incubation experiments are, at present, one of few methods to provide insights into the mechanisms of polysaccharide processing—extracellular hydrolysis and selfish uptake—with the possibility to link the function to the identity of specific bacteria: to date, selfish uptake cannot be detected solely via 'omic analyses (29). Here, we present a simple approach to detect bacteria—in pure cultures and complex environments—that are actively taking up polysaccharides through a selfish mechanism (Fig. 2). In brief, the FLAPS of interest is added to a liquid sample or culture medium and incubated, and subsamples are periodically collected and filtered. Selfish substrate accumulation in the periplasm can be visualized with a standard epifluorescence microscope after initial DNA staining (e.g., DAPI) through co-localization

of the FLAPS signal and the nucleic acid counter stain. FLAPS signals without a nucleic acid counterstain should be excluded as background noise. This simple and straightforward approach allows simultaneous quantification of total and selfish cells and answers the first key question: are selfish bacteria that use this specific polysaccharide active in my sample? This experimental setup also permits further investigations: the same filter can be used to analyze bacterial community composition by 16S rRNA amplicon sequencing (Fig. S1D). Moreover, collecting the filtrate allows measurement of extracellular hydrolysis rates, using gel permeation chromatography and fluorescence detection (Fig. 1E; Fig. S1A) (24).

A wide range of soluble and semi-soluble polysaccharides can be labeled to probe a range of polysaccharide metabolisms (Table S1, see Online Methods for FLAPS production procedure). FLAPS have been successfully used with diverse pure cultures from different phyla and a range of different environmental microbiomes, including marine (seawater and sediments [26, 27, 30, 31, 35, 36, 47, 48]), limnic and riverine (49–52), and rumen (cattle and sheep 13, 29, 53) (Table S1).

## Measuring extracellular hydrolysis and selfish uptake of FLAPS—an environmental example

Using the procedures described above, we incubated an environmental sample—surface seawater collected in September off Helgoland (North Sea)—with three FLAPS (laminarin, xylan, and chondroitin sulfate) to determine the relative contributions of selfish bacteria and external hydrolyzers to polysaccharide degradation. External hydrolysis of polysaccharides, which produces low molecular weight hydrolysis products in the surrounding medium, was measured via gel permeation chromatographic analysis of the filtrate collected at each time point. Hydrolysis rates were calculated based on the shift in molecular weight classes as a polysaccharide is systematically hydrolyzed to lower molecular weight hydrolysis products over time (Fig. 1E).

The incubations showed rapid selfish uptake of both laminarin and xylan: 16% and 6% of cells were stained by these two FLAPS, respectively, already at the initial (t = 1 h) time point (Fig. 3A). Selfish uptake increased up to 72 h; a low number of chondroitin-stained cells were also detected. All three polysaccharides were also externally hydrolyzed, with high hydrolysis rates of chondroitin and xylan detected in the 48 and 72 h samples, respectively (Fig. 3B). Total cell counts increased from $0.9 \times 10^9$ cells $L^{-1}$ to $1.5 \times 10^9$ cells $L^{-1}$ in all incubations (amended and unamended) within 24 h of incubation, but then diverged, with cell counts in the chondroitin incubations increasing to ca $3 \times 10^9$ cells $L^{-1}$ at 72 h, a smaller increase in the xylan incubations, and a decrease in the laminarin and unamended incubations (Fig. 3C). Note that the concentrations of FLAPS added to seawater samples (3.5 µmol monomer-equivalent per liter of seawater) is aimed are producing detectable signals during gel permeation chromatography/fluorescence detection—in other words, to obtain a rate of extracellular hydrolysis—and representrepresent low addition of organic carbon (typically on the order of 20 umol C $L^{-1}$ seawater), especially considering typical organic carbon concentrations in the surface ocean and nearshore environments. Thus, the FLAPS are not intended as a growth substrate, although an increase in cellular abundance is sometimes observed, as in the current incubation for chondroitin and xylan. By the same measure, the fact that the incubations are carried out in unfiltered seawater means that grazers and viruses are present in the same incubations as the bacteria. A decrease in cell numbers during incubation, as observed in the laminarin and unamended incubations, can be due to the activities of grazers and/or viruses naturally present in seawater. In any case, counting total and selfish cells clearly answers the first key question: selfish bacteria were present and actively taking up all three FLAPS, but selfish uptake of laminarin and xylan was greater than for chondroitin.

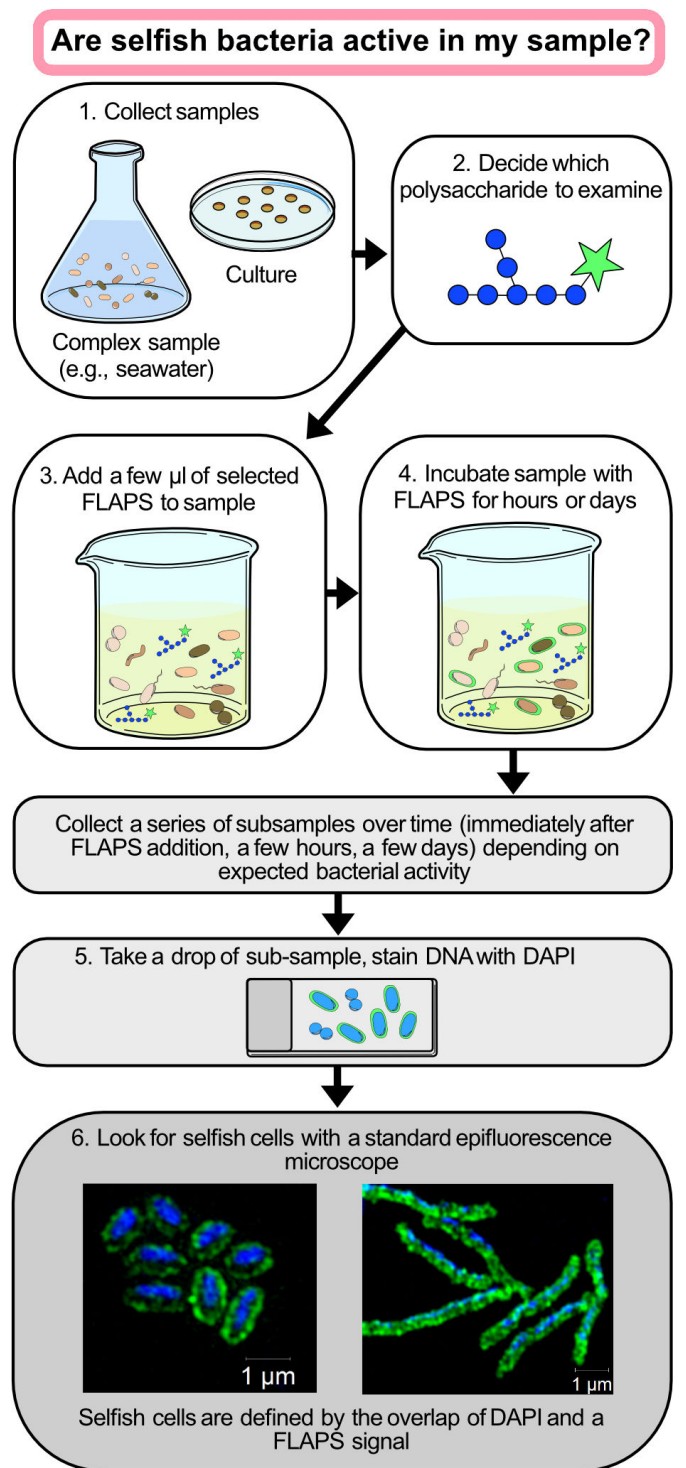

**FIG 2** Simple workflow for the identification of selfish bacteria in pure cultures or complex samples through a fluorescently labeled polysaccharide (FLAPS) incubation experiment.

## Sequencing to gain insight into complex communities

After discovering whether selfish bacteria are present in a complex community, further questions may relate to bacterial identity: which organisms are present in the initial sample? To what extent does the community change with increasing incubation time? Are there any indications of specific organisms responding to a FLAPS amendment?

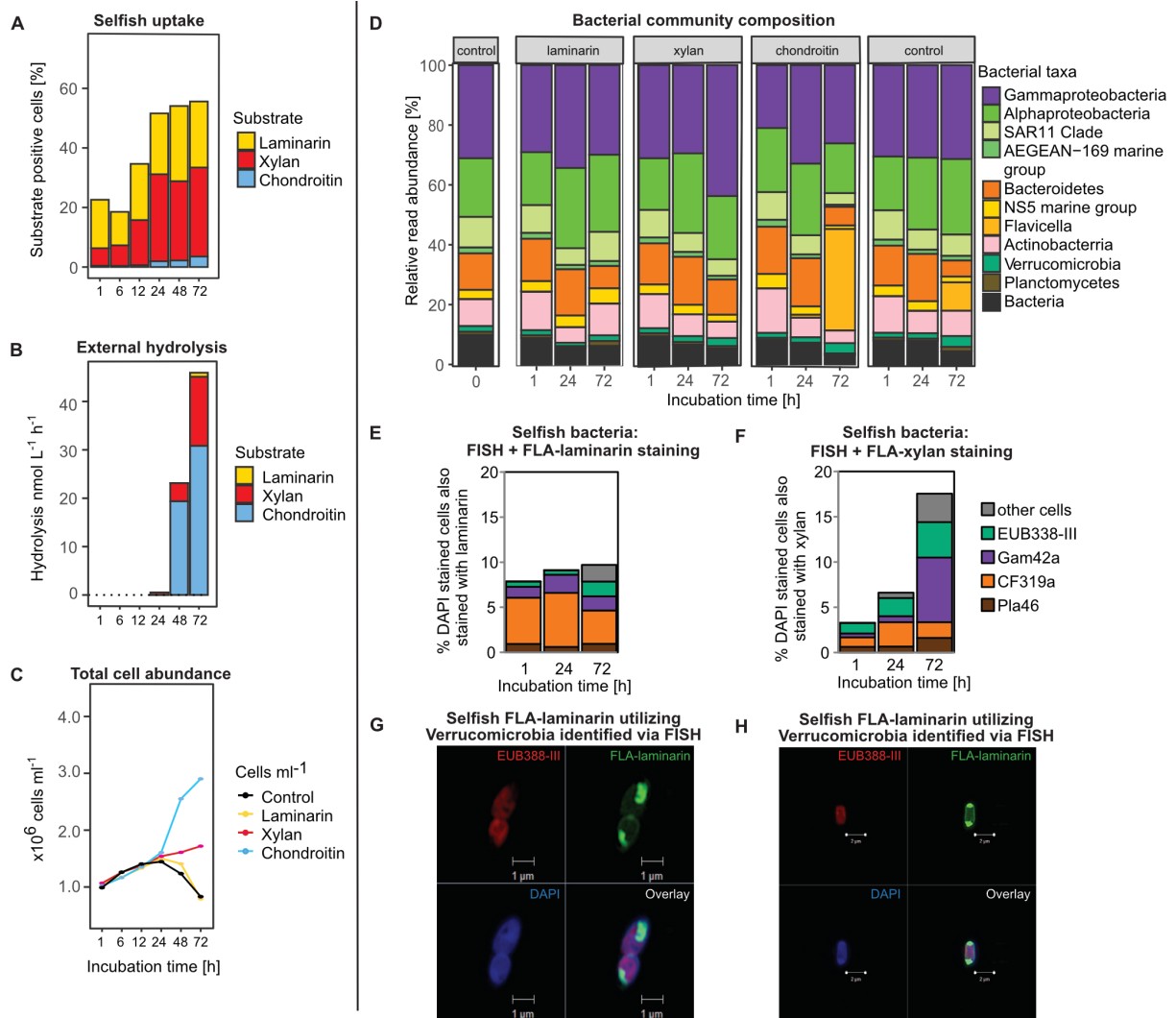

**FIG 3** Polysaccharide utilization pattern in surface water from Helgoland in September over the course of a 72 h incubation. (A) Laminarin and xylan uptake stained a high proportion of cells already from the beginning of the incubation, whereas (B) extracellular hydrolysis was comparatively rapid for xylan and chondroitin sulfate. (C) Microbial cell counts developed differently by substrate after the initial 24 h incubation. The (D) incubation-dependent changes of the initial community composition over 72 h show an FLA-chondroitin-dependent increase in *Flavicella* reads, but otherwise, the community composition remained unchanged. Taxonomic identification with FISH showed (E) a large contribution of *Bacteroidota* (CF319a) to selfish laminarin uptake, whereas (F) a more diverse array of organisms including *Gammaproteobacteria* (Gam42a) and *Verrucomicrobiota* (EUB338-III) took up xylan. (G + H) Super-resolution structured illumination images showing different polysaccharide accumulation patterns after 24 h incubation in FLA-laminarin stained cells (green), counterstained with DAPI (blue), and taxonomic correlation with FISH probes (red).

In these cases, amplicon sequencing of 16S rRNA genes (next-generation sequencing; NGS) can be the next step. In this case, unamended treatment controls (incubations to which no FLAPS were added) should be sequenced for the same time points as FLAPS-amended incubations to distinguish bottle effects from any substrate-dependent community responses. The correlation of a substrate-dependent change in bacterial taxa with a change in polysaccharide utilization can help identify potential taxa involved in the process (22). Furthermore, selfish organisms can be phylogenetically stained and counted microscopically using fluorescence *in situ* hybridization (FISH) (26). Taxonomically specific FISH probes can be selected (or designed) to confirm the absolute abundance of a bacterial group and create a direct, visual link to selfish polysaccharide accumulation (Fig. 3G).

The initial bacterial community in Helgoland waters in September was composed of *Gamma-* and *Alpha-proteobacteria*, *Bacteroidota,* and *Actinobacteria* (Fig. 3D). Over the course of incubations, minor changes in abundance occurred within these groups. However, the bacteroidetal *Flavicella* was an exception*,* showing a large increase—of 34% and 10%—at 72 h in the chondroitin and control incubations, respectively, compared with the initial community.

## Revealing links between function and taxonomy—FISH on FLAPS-stained cells

Combining FLAPS uptake with FISH links an organism directly with its substrate, yielding information that is otherwise extremely difficult or impossible to obtain, particularly from environmental samples. Different FISH methods targeting rRNA can be used to visualize bacterial groups. Extensive testing (see Methods) has demonstrated that modifying the protocol of Manz et al. (1992) (45) using quadruple-labeled oligonucleotide probes is most suitable for identifying FLAPS-stained selfish bacteria. It is compatible with FLAPS incubation because the procedure is less harsh when compared with other protocols (i.e., CARD-FISH see Methods), has fewer steps, and gives a detectable FISH signal even for small cells from environmental samples.

Since the Helgoland incubations showed high selfish uptake of laminarin and xylan (Fig. 3A), we focused our FISH investigations on these samples, using probes targeting the abundant *Bacteroidota* (CF319a) and *Gammaproteobacteria* (GAM42a) as well as the minor phyla *Verrucomicrobiota* (EUB338-III) and *Planctomycetes* (PLA46), which have previously been found to be capable of selfish uptake (30, 31, 54). Laminarin incubations were clearly dominated by selfish *Bacteroidota,* especially during the initial 24 h (Fig. 3E), whereas in the xylan incubations, selfish *Gammaproteobacteria* and also *Verrucomicrobiota* increased in proportion especially by 72 h of incubation (Fig. 3F). We note, moreover, that the numbers of FISH- and FLAPS-positive cells are likely underestimated because the FISH procedure can lead to a loss of substrate signal in cells (Fig. S2; see Materials and Methods).

## Super-resolution microscopy—visualization of individual selfish substrate accumulation patterns

In addition to standard epifluorescence microscopy, high-resolution visualization of the accumulated FLAPS within the cell can be carried out using super-resolution structured illumination microscopy (SR-SIM) (Fig. 3G). Cells from the Helgoland FLA-laminarin incubation were identified as members of the *Verrucomicrobiota* by FISH (Fig. 3G) and showed two different versions of polar substrate accumulation pattern with an enlarged periplasmic space, in contrast to elongated cells identified as *Gammaproteobacteria* that stained more evenly among the periplasm (Fig. 2). To date, cell staining has shown two distinct patterns. Halo-like staining within the entire periplasmic space shows an even signal seen in *Gammaproteobacteria* and *Bacteroidota*. Polar staining with one or both ends of the cell showing a clear increased signal, sometimes associated with an enlargement of the periplasmic space, observed in *Verrucomicrobiota* and *Plantomycetes* (30). The use of a membrane stain can show the co-localization of the polysaccharide-associated green fluoresceinamine signal with the red membrane stain Nile Red in a fluorescent intensity line grating (30), further demonstrating polysaccharide uptake into the periplasmic space.

In addition, visualizing selfish uptake can reveal cellular variability in substrate processing as shown in a pure culture of *Bacteroides thetaiotaomicron,* a bacterium of the human gastrointestinal tract (Fig. 4A). When *B. thetaiotaomicron is* incubated with FLA-yeast mannan (Fig. 4A), individual cells in bacterial cultures can exhibit a variable extent of staining. In some instances, all cells in a pure culture take up FLAPS to a similar extent, whereas in other instances, some cells are more strongly stained than others. Identifying and monitoring possible differences in staining among cells is essential in applications where cells are used for bioprocesses (e.g., production of fuels such

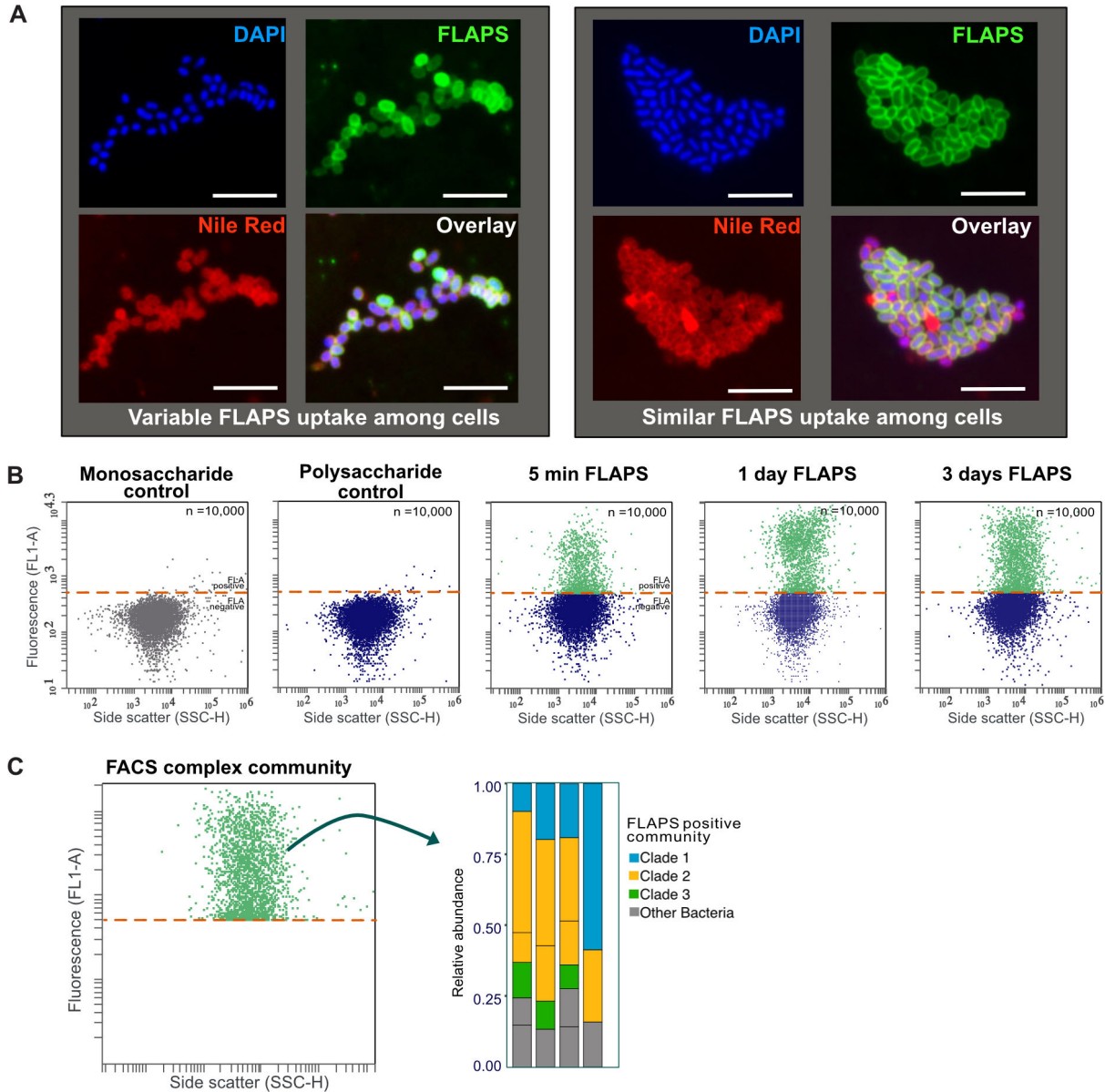

**FIG 4** Metabolic phenotyping and flow cytometry of cultures of human gut microorganisms and of complex communities. (A) Visualization of cell-to-cell variability in the extent of staining of strains of *Bacteroides thetaiotaomicron* incubated with fluorescently labeled yeast mannan. Note how the FLAPS staining is variable among individual cells in the left panel, whereas the FLAPS staining is similar among all cells in the right panel. The cell DNA is shown by DAPI staining in blue, the FLAPS is shown in green, and cell membranes are shown by Nile red staining in red. Scale bar = 5 μm (B) Quantification of FLAPS uptake into *Bacteroidetes thetaiotaomicron* by flow cytometry. Shown in the first two panels are negative controls: *Bacteroidetes thetaiotaomicron* grew on unlabeled monosaccharides and unlabeled polysaccharides. The subsequent three panels show the change in fluorescence of the cells with incubation in FLAPS for 5 min, 1 day, and 3 days. Data revisualized from Klassen et al., 2021 for comparison and schematic purposes. The red dotted line represents the fluorescence threshold of the control community. (C) Schematic representation of fluorescence-activated cell sorting of FLAPS-positive cells in combination with sequencing.

as ethanol, butanol, fatty acid derivatives, or natural products), as it affects biosynthesis performance, specifically enzyme activity, or expression level (55). Mechanisms underlying microbial cell-to-cell variability in staining that are not based on genotype are not well understood, but FLAPS incubation can visualize such variability.

## Flow cytometry: tracking specific organisms

Especially for cases in which identification of specific cells is difficult, or—for pure cultures, for example—when the relative change in FLAPS uptake needs to be measured over short timescales, flow cytometry can be helpful. Bacteria that take up FLAPS can be flow cytometrically identified based on their physical and fluorescence properties within minutes and sorted based on the fluorescence signature of FLAPS accumulation (Fig. 4B and C) (32, 33, 56, 57). Flow cytometry can be used in environmental samples to identify subpopulations of FLAPS-stained cells. Various controls are required, including a blank control of the community with the nucleic acid counterstain for background noise calibration (Fig. S3A). Since flow cytometry of killed controls is problematic (see Methods), a possibility for a negative control is the addition of FLAPS to a fixed and thus inactivated sample to account for any unspecific binding of the substrate (Fig. S3C through I). For pure cultures, flow cytometry can be used to identify differences in uptake efficiency between cultures by plotting fluorescence intensity over forward scatter (a proxy for cell size) or side scatter (a proxy for cell granularity) (Fig. 4B) (29). The pure culture without FLAPS, as well as the medium without cells but with FLAPS, should be used to calibrate the background noise.

Fluorescence-activated cell sorting (FACS) of selfish populations can be used to assess the taxonomic composition and functional potential of active selfish organisms in a complex community. Here, selected bacterial populations are separated from the sample and enriched. Applying FISH on sorted cells can quantify taxa in a sorted population and link taxonomy to uptake patterns (35, 53, 57). Moreover, cells stained at different intensities with the FLAPS could be used as a proxy to differentiate between different populations within a sample. However, in environmental samples, quantification of unstained vs. stained cells would only be a rough estimate, as there are potentially very pronounced differences in the staining pattern among different taxa at a given time (Fig. 3G). Note in all of these cases that microscopic validation of selected populations after sorting is necessary and advised. Additionally, a nucleic acid stain can be used as an independent parameter to ensure that cells (and not background signals) are sorted.

## Conclusions

Several major points emerge from our investigations of polysaccharide processing: most importantly, by overlooking selfish bacteria, a major substrate processing mechanism carried out by bacteria in a wide range of environments is missed. We note that external hydrolysis of laminarin was minimal in our incubations; however, selfish uptake was rapid, even in the initial community collected from the ocean (Fig. 3A through C). However, low selfish uptake of chondroitin shows that external hydrolysis can also be important—and that the importance of the polysaccharide processing mechanism varies by substrate, since the same starting communities were present in each incubation. Moreover, the ability to carry out selfish uptake is phylogenetically widespread, as indicated by the fact that in the laminarin and xylan incubations, substantial selfish uptake did not correlate with changes in specific taxa in the bulk community analysis. This observation—and the broad range of selfish cells, especially in the xylan incubation at 72 h—suggests the widespread prevalence of the selfish mechanism among diverse bacteria. Although selfish *Bacteroidota* and *Verrucomicrobiota* constitute a major portion of the total in early xylan incubations, *Gammaproteobacteria* constitute a large fraction of selfish bacteria at 72 h. Therefore, to investigate the processing of polysaccharides by microorganisms and microbial communities, phenotypic approaches that allow for *in situ* probing are essential (56).

## ACKNOWLEDGMENTS

We thank our colleagues from Biologische Anstalt Helgoland, especially Antje Wichels, Eva-Maria Brodte, and Uwe Nettelmann, for enabling the sampling campaign on Helgoland and facilitating substrate incubations and sample preparation. In addition,

we thank the crew of the *Aade* for sample collection, Maria Belen Gonzalez Pino for help with incubation experiments, sample preparation, and microscopic analysis, and Sherif Ghobrial (UNC) for help with the work on the fluorescently labeled polysaccharides.

G.R. has received funding from the European Union's Horizon 2020 research and innovation program under the Marie Sklodowska-Curie grant agreement No. 840804 and the Deutsche Forschungsgemeinschaft (DFG, German Research Foundation) Project number 496342779. This study was supported and funded by the Max Planck Society and the U.S. National Science Foundation (OCE-1736772 and −2022952 to CA).

All authors conceived different aspects of the experimental study; G.G. performed sample collection, G.G. conducted FLAPS incubations; G.G. and G.R. performed FISH, flow cytometry, and microscopy analyses; C.A. synthesized FLAPS and analyzed samples for external hydrolysis; G.G., C.A., and G.R. analyzed the data; R.A., C.A., and G.R. secured funding; G.G., G.R., and C.A. produced the figures and tables; all authors contributed to writing the manuscript.

## AUTHOR AFFILIATIONS

[1]Department of Molecular Ecology, Max Planck Institute for Marine Microbiology, Bremen, Germany

[2]Microbial-Carbohydrate Interactions Group, Faculty of Biology/Chemistry, University of Bremen, Bremen, Germany

[3]Department of Earth, Marine, and Environmental Sciences, University of North Carolina-Chapel Hill, Chapel Hill, North Carolina, USA

## AUTHOR ORCIDs

G. Reintjes http://orcid.org/0000-0001-7085-4683
B. M. Fuchs http://orcid.org/0000-0001-9828-1290
C. Arnosti http://orcid.org/0000-0002-6074-5341

## FUNDING

| Funder | Grant(s) | Author(s) |
|---|---|---|
| Deutsche Forschungsgemeinschaft | 496342779 | G. Reintjes |
| Horizon 2020 Framework Programme | 840804 | G. Reintjes |
| National Science Foundation | OCE-1736772 | C. Arnosti |
| National Science Foundation | OCE-2022952 | C. Arnosti |
| Max-Planck-Gesellschaft | | B. M. Fuchs |
| | | R. Amann |

## AUTHOR CONTRIBUTIONS

G. Reintjes, Conceptualization, Data curation, Formal analysis, Funding acquisition, Validation, Visualization, Writing – original draft, Writing – review and editing | G. Giljan, Conceptualization, Data curation, Formal analysis, Investigation, Methodology, Software, Visualization, Writing – original draft | B. M. Fuchs, Conceptualization, Funding acquisition, Investigation, Resources, Software, Supervision, Visualization, Writing – original draft | C. Arnosti, Conceptualization, Funding acquisition, Supervision, Validation, Visualization, Writing – original draft, Writing – review and editing | R. Amann, Conceptualization, Funding acquisition, Project administration, Resources, Supervision, Writing – original draft, Writing – review and editing

## ADDITIONAL FILES

The following material is available online.

## Supplemental Material

**Supplemental figures and tables (Spectrum01602-24-S0001.pdf).** Figures S1 to S5 and Tables S1 to S4.

**Supplemental material (Spectrum01602-24-S0002.docx).** Reference list of all cited publications in Table S1.

## Open Peer Review

**PEER REVIEW HISTORY (review-history.pdf).** An accounting of the reviewer comments and feedback.

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
