## [Reviewer comments · Microbiology Spectrum]

Microbiology Spectrum

Using phenotyping to visualize and identify selfish bacteria: a methods guide

Greta Reintjes, Greta Giljan, Bernhard Fuchs, Carol Arnosti, and Rudolf Amann

Corresponding Author(s): Greta Reintjes, Universitat Bremen

Review Timeline:

Submission Date:	July 1, 2024
Editorial Decision:	October 14, 2024
Revision Received:	March 5, 2025
Editorial Decision:	April 28, 2025
Revision Received:	May 6, 2025
Accepted:	May 20, 2025

Editor: Sandi Orlic

Reviewer(s): Disclosure of reviewer identity is with reference to reviewer comments included in decision letter(s). The following individuals involved in review of your submission have agreed to reveal their identity: Mirjam Czjzek (Reviewer #4)

Transaction Report:

DOI: <https://doi.org/10.1128/spectrum.01602-24>

Re: Spectrum01602-24 (Using phenotyping to visualize and identify selfish bacteria: a hunting guide)

Dear Dr. Greta Reintjes:

Thank you for the privilege of reviewing your work. Below you will find my comments, instructions from the Spectrum editorial office, and the reviewer comments.

Revision Guidelines

Sincerely,
Sandi Orlic
Editor
Microbiology Spectrum

Reviewer #2 (Comments for the Author):

Summary:

This manuscript presents a workflow for visualizing and identifying heterotrophic bacteria in pure culture or environmental samples that carry out "selfish" polysaccharide degradation. This selfish uptake mechanism involves polysaccharides binding to the outer membrane of cells and being partially hydrolyzed in the periplasmic space to prevent "cheaters" from benefiting from cellular investment in hydrolytic enzymes. The study presents data from an environmental seawater sample and demonstrates

that polysaccharide mechanisms varied by substrate, and that selfish cells may be phylogenetically diverse and widespread in this environment. The manuscript also presents details on the various components of the workflow, including how to differentiate selfish cells from background signal.

I believe the workflow, seawater data, and discussion of background signal presented in this manuscript are interesting and would be helpful to other researchers. The manuscript is well written, and the figures are generally clear. However, I am not convinced that the authors' major conclusion about the occurrence of selfish uptake in the seawater sample is supported by their data. I think clearer, more zoomed-in microscope images are needed to support this conclusion.

Major comments:

1) Figure 2 Step 6, Figure 3G, Figure 4A, lines 187-190, and various places in manuscript text: Based on the data presented in the microscopic images of the seawater sample, I am not convinced that the selfish uptake mechanism is happening here. Specifically, the images do not clearly demonstrate uptake of labeled polysaccharides (FLAPS) into the bacterial periplasmic space. In my opinion, the Nile Red membrane stain mentioned in lines 187-190 and shown in Figure 4A is what the authors really need to demonstrate selfish uptake, but it has not been done convincingly here. In Figure 2 Step 6, how do you know that the FLAPS are not simply binding to the outside of the cells (and perhaps being degraded into small products that are released into the growth medium)? It would be helpful to show microscope images of non-selfish cells for comparison. In Figure 4A, the images in the left panel are difficult to resolve at the individual cell level. The right panel is clearer, but I suggest zooming in on individual cells to show co-localization of FLAPS and membrane stains. You need to show FLAPS in the periplasm between the outer and inner membranes. This is critical to establishing the presence of selfish uptake in this sample, which is a major objective of the study.

2) I think the 2 major strengths of this manuscript are the seawater analysis (Figure 3 & 4A) and the tips on distinguishing background signal from selfish cells (discussed at length in the Supplementary Information), which appear to be very important for the accuracy of this workflow while also being quite complex and sample-specific. I suggest the authors consider moving some of the background signal discussion into the main paper and figures, and also provide more microscopic images of fluorescent background (non-cellular) particles to compare to FLAPS-labeled cells, with the goal of helping other researchers distinguish them. The data in Figure S3B-C clearly show a difference between selfish cells and background signal, but the background signal is very high and sample-specific. Would researchers need to perform this comparison for every sample? The images in Figure S5 are helpful, but the manuscript could be improved with zoomed-in images of individual FLAPS-labeled cells compared to non-cellular fluorescent background particles. The resolution in the Figure 2 Step 6 image is closer to what I think you need here. The background signal discussion makes me think that the workflow presented in this manuscript is not "simple" as stated in the Figure 2 caption, but rather something that must be done with care and precision (see lines 307-308). That is fine, but I think the manuscript could emphasize this further and make the helpful tips in the Supp Info more prominent, as they are non-trivial. This is particularly important in light of my previous major comment about distinguishing FLAPS simply bound to organic debris (or even cells) rather than actually taken up into cellular periplasm.

Minor comments:

- 1) General: Please clarify throughout the manuscript that the selfish uptake mechanism described herein only applies to Gram negative bacteria (i.e., cells that contain an outer membrane and periplasmic space). Also please comment on whether Gram positive or Gram resistant bacteria can undergo selfish uptake (perhaps through a different mechanism?) and whether the methods in this manuscript can detect that. If not, please discuss the limitations in investigating mixed community samples.
- 2) General (figures): Please give each panel its own letter to reduce confusion. For example, split Figure 3G into G (left image) and H (right image).
- 3) Lines 45-46: Please add that fungi also contribute substantially to degradation of plant polysaccharides in the environment (i.e., cellulose degradation), with appropriate references.
- 4) Figure 1D: Please explain this image in more detail. Based on reading the text, I assume these are individual cells with DAPI in blue and FLAPS in green surrounding the cell. This was not immediately clear from the figure caption. (I didn't know if this was something inside an individual cell, outside an individual cell, etc.)
- 5) Figure 1E: Please clarify in the figure caption that this represents the external hydrolysis mechanism, not selfish uptake. Polysaccharides are being degraded into smaller products outside the cells, in the filtrate.
- 6) Lines 91-93: Please specifically explain which parts of this manuscript are new compared to the previous Giljan et al. 2022 paper.
- 7) Line 93: Please clarify which data in this study are on human gut microorganisms. Is it the flow cytometry data in Figure 4B-C? If so, please clarify in the text and figure caption.
- 8) Line 169: Please reference Figure 3A.
- 9) Line 185-187 & Figure 3G: I'm not sure I see an enlarged periplasmic space in either of these images. Please clarify. Both appear to show polar localization. Also, you need a way to prove that the FLAPS are indeed located in the periplasmic space (see my major comment about this).
- 10) Figure 3C: It appears that cells started dying in the control and laminarin samples, but continued to grow in the chondroitin samples. Does this imply that selfish uptake does not support population growth as well as external hydrolysis (compare Figure 3C to 3A & 3B)? If so, please comment on the relative importance of these 2 mechanisms in this environment.

- 11) Figure 3 caption: Please quantify the terms "increase," "unchanged," etc.
- 12) Lines 192-194 & Figure 4A: Please describe specifically how the images show heterogeneous vs. homogeneous uptake patterns, and exactly what these terms mean in this context. The images are difficult to interpret.
- 13) Figure 4B: Please explain the significance of the red dotted lines.
- 14) Figure 4C: This is a conceptual model. Can you perform this analysis on your seawater sample and show real data to demonstrate the utility of this method (combining flow cytometry with bacterial community profiling)?
- 15) Line 220-221 & Figure 3G: Please specify the time difference between the 2 panels, either here or in the figure caption.
- 16) Lines 226-241: These conclusions are very interesting and generally well supported by your seawater data (although see my first major comment).
- 17) Lines 242-262: I don't think this text is needed, as it does not describe conclusions from this particular study. The authors could move it to the Introduction if desired.

Reviewer #3 (Comments for the Author):

General comment - The authors have not presented data that shows "selfish" uptake, which infers nutrient acquisition by denying others access to the substrate. To accurately show a selfish mechanism the authors would need to have competition assays where two bacterial strains that can use a substrate do not get equal access to substrate. What they have done is develop a method to identify bacteria that can use uptake labeled polysaccharides and be able to visualize them amongst a complex community. It is still a useful tool, but the FLAPS data as presented does not identify selfish bacteria, just those that can import a tagged substrate. Instead of trying to sell a story about identifying selfish bacteria, the authors should reframe the manuscript as a methods guide to using the FLAPS and highlight the various applications, but also add context about all of the extra instrumentation required to use FLAPS effectively (e.g. FISH, microscopy, FACS, etc.).

p. 3 l. 121 - Why are the limnic and gut microbiome data not available in this paper? If the authors are mentioning them, they should be included. The authors have a sizable Supplemental Data document, so what is the rationale for mentioning, but then not sharing, those data? Nearly all of the unpublished data in Table S1 are for *B. thetaiotaomicron*, are the authors expecting to publish these results later, and if so, why mention them now?

p. 4 l. 124-125 - What is the rationale for the substrate choice (laminarin, xylan, and chondroitin)? The environmental sample is surface sea water, so why have one substrate that is more often considered terrestrial-plant derived? Are marine plants rich in xylan? If you wanted to use marine-plant specific substrates, wouldn't alginate or agarose be better choices?

p.5 l. 136 - Why is cell growth so poor? The authors say cell counts increased, but there is not even a doubling of cell number within 24 hours and is barely 3 cell doublings in 72 hours. Are FLAPS even viable substrates for growth? How much label is attached to a polysaccharide and how does that impact degradability?

Figure 3G - Why is the FLA-laminarin signal just at the poles of the bacteria? If the probe was periplasmic, shouldn't it be a ring as is seen in Figure 1D?

Figure 4. There is no mention at all of Nile Red staining in the text. The authors need to explain why it was done and what the results mean.

Supplementary Methods - As written there is not enough information for another researcher to generate FLAPS. This is just a rewording of the short description on p.8 l.280-285. Furthermore, there is a mention of unpublished results in the description to generate FLAPS. This also occurs in several other places in the manuscript. There is no reason to not include the missing information so the authors should include it.

Subject: Spectrum01602-24 Decision Letter

Re: Spectrum01602-24 (Using phenotyping to visualise and identify selfish bacteria: a hunting guide)

Dear Editor,

We thank the reviewers for their constructive comments and have addressed them in detail below. *Our responses are in italics and blue font.* We have used Track Changes in the manuscript to highlight the revisions that we have made.

Reviewer #2 (Comments for the Author):

Summary:

This manuscript presents a workflow for visualizing and identifying heterotrophic bacteria in pure culture or environmental samples that carry out "selfish" polysaccharide degradation. This selfish uptake mechanism involves polysaccharides binding to the outer membrane of cells and being partially hydrolyzed in the periplasmic space to prevent "cheaters" from benefiting from cellular investment in hydrolytic enzymes. The study presents data from an environmental seawater sample and demonstrates that polysaccharide mechanisms varied by substrate, and that selfish cells may be phylogenetically diverse and widespread in this environment. The manuscript also presents details on the various components of the workflow, including how to differentiate selfish cells from background signal.

I believe the workflow, seawater data, and discussion of background signal presented in this manuscript are interesting and would be helpful to other researchers. The manuscript is well written, and the figures are generally clear. However, I am not convinced that the authors' major conclusion about the occurrence of selfish uptake in the seawater sample is supported by their data. I think clearer, more zoomed-in microscope images are needed to support this conclusion.

Major comments:

1) Figure 2 Step 6, Figure 3G, Figure 4A, lines 187-190, and various places in manuscript text: Based on the data presented in the microscopic images of the seawater sample, I am not convinced that the selfish uptake mechanism is happening here. Specifically, the images do not clearly demonstrate uptake of labeled polysaccharides (FLAPS) into the bacterial periplasmic space. In my opinion, the Nile Red membrane stain mentioned in lines 187-190 and shown in Figure 4A is what the authors really need to demonstrate selfish uptake, but it has not been done convincingly here. In Figure 2 Step 6, how do you know that the FLAPS are not simply binding to the outside of the cells (and perhaps being degraded into small products that are released into the growth medium)? It would be helpful to show microscope images of non-selfish cells for comparison. In Figure 4A, the images in the left panel are difficult to resolve at the individual cell level. The right panel is clearer, but I suggest zooming in on individual cells to show co-localization of FLAPS and membrane stains. You need to show FLAPS in the periplasm between the outer and inner membranes. This is critical to establishing the presence of selfish uptake in this sample, which is a major objective of the study.

The reviewer raises the question of whether 'selfish uptake' can occur. In the eight years since we published our first manuscript about selfish uptake in the ocean (Reintjes et al. 2017) and ten years since Cuskin et al. (2015) published the first manuscript about selfish uptake in a gut bacterium, we have dealt extensively with this question. The principal objective of this

manuscript is to provide an easy-to-follow guide for a wider range of scientists to identify selfish bacteria which may be active in a diverse range of environments. We do not believe that all scientists must use super-resolution structured illumination microscopy to achieve this goal.

To these points:

-We note that we can distinguish between selfish uptake and external hydrolysis, as discussed in the manuscript, because we also sample the seawater in which bacteria grow: external (extracellular) hydrolysis of polysaccharides leads to production of lower molecular weight hydrolysis products, which are detected by gel permeation chromatography and fluorescence detection (see Fig. 1E). If bacteria were merely binding FLAPS to their cell surface and hydrolyzing them, then the hydrolysis products would escape into seawater, and be detected via gel permeation chromatography/fluorescence detection. This is a step that can be carried out in a relatively low-cost manner and which a broad range of scientists without access to high-tech equipment can do. (Selfish uptake is not always accompanied by external hydrolysis, however.)

-As discussed in the manuscript, we have previously published high-resolution microscopic images using super-resolution structured illumination microscopy. For example, we reproduce below Fig. S8 from Reintjes et al. 2017, demonstrating staining by DAPI, Nile red, and FLAPS. This information was already referenced in the manuscript.

- The request to show the FLAPS between a gram-negative bacterium's inner and outer membrane awaits further methodological improvements in the field: at best, our resolution with super-resolution microscopy is 20nm. The periplasm is approximately 7.5nm thick; even the best super-resolution images cannot resolve this.

-Note that Fig. 2, step 6 demonstrates what selfish bacteria look like under a standard epifluorescence microscope.

Reintjes et al. Supplementary Figure S8

Fig. S8, from Reintjes et al. (2017) showing SR-SIM images of halo-like substrate staining (FLA-laminarin; green) in the Northern Temperate station after 6 days incubation. Scale bar is 0.5 μ m.

Further information about determining selfish uptake:

All incubations are conducted with a negative control sample, a water sample incubated without FLAPS for the same duration of time. In this sample, we monitor the communities' autofluorescence. We do this because if the threshold for FLAPS cell counting is set too low, the autofluorescence signal can be misinterpreted as a FLAPS signal. To avoid this, we have the control samples and deliberately set our threshold higher than any autofluorescence signal. We have not included the images, as they are predominantly blank. Still, we added to the manuscript the values of our thresholds in exposure time and signal background ratio in the images to our methods section.

“Total DAPI counts and selfish polysaccharide uptake were analyzed using a fully automated epifluorescence microscope (Zeiss AxioImager.Z2, Carl Zeiss), equipped with a with a cooled charged-coupled-device (CCD) camera (AxioCam MRm; Carl Zeiss) and a Colibri LED light source (Carl Zeiss) with three light-emitting diodes (UV-emitting LED, 365 ± 4.5 nm for DAPI; blue emitting LED, 470 ± 14 nm for FLA-PS 488; red-emitting LED, 590 ± 17.5 nm for the tyramide Alexa 594, FISH), combined with the HE-62 multifilter module (Carl Zeiss). This module consists of a triple emission filter TBP 425 (± 25), 527 (± 27), LP 615, including a triple beam splitter of TFT 395/495/610. Images were acquired using a 63X oil immersion plan apochromatic objective with a numerical aperture of 1.4 (Zeiss) at the selected wavelengths specified above. Images from the substrate channel (FLAPS, 470nm) were acquired at three exposure times (10 ms, 35 ms, and 140 ms) to cover the diversity of signal intensities and patterns of FLAPS accumulation. A minimum of 45 fields of views were acquired, microscopic images exported into the modified image analysis software ACMEtool (M. Zeder, Technology GmbH, <http://www.technobiology.ch> and Max Planck Institute for Marine Microbiology, Bremen), and signals evaluated (Supplementary Table S2) according to Bennke et al. (2016)⁴⁰. Briefly, a positive signal in DAPI and FLAPS images with a minimum overlap of 30%, as well as a minimum signal background ratio of 1, were required for positive identification as a FLAPS-stained cell. An overlap of all three signals indicated a FLAPS-stained cell identified by a specific FISH probe (see below).” (Line 403 - 420)

If there is an autofluorescence signal higher than our threshold, for example, due to cyanobacteria, we perform a further control. We take an additional microscope image called AUTO. This image is taken with red-emitting LED filters and 590 ± 17.5 nm exposure times. Any selfish cells that also have auto signals are not counted. The explanation is given below:

„Each image was also recorded on the Auto signal (590 nm) with an exposure time of 300ms to visualize potential autofluorescence from pigments within cells. All bacteria that had a positive signal on the Auto, DAPI and FLAPS channels were recognized as Cyanobacteria and excluded from the calculations for selfish uptake.” (Line 421 - 425)

Further comments regarding the reviewer's comment on the unspecific binding of FLAPS:

FLAPS have a labelling density of ~ 1 . The signal intensity from detection with routine epifluorescence microscopy is at least 400 - 1000 fluorophores. Therefore, every selfish cell would have to be covered by more than 1000 sugar chains to show the signal we see. This is likely to have an effect on cell functioning through obstruction of critical cell surface structures

and affect nutrient transport and cell signalling. A thick sugar layer can also impede the diffusion of small molecules to the cell surface. In our pure culture experiments, we do not see an adverse effect, but rather a positive impact, of FLAPS on growth. Therefore, we do not believe FLAPS bind to the cell surface unspecifically.

To critically evaluate the point that the staining is indeed intracellular and not just specific or unspecific surface binding, we have performed several tests, which we highlight here: In a previous publication (Reintjes et al. 2017; referenced in the manuscript), we performed a super-resolution microscope study of FLAPs-stained cells and showed that the signal is co-localised with the cell membrane. The figure below from Reintjes et al. 2017 shows bacterial cells stained with FLA-PS (green), Nile red (red; membrane) and DAPI (blue; DNA). White arrows indicate sections along which the fluorescence intensity line profiles were recorded. Scale bars=1 μ m. Corresponding profiles indicating co-localisation of substrate and membrane are shown on the right. The first is a *Bacteroidota* cell from the Northern Temperate station stained with FLA-laminarin. The second is a *Bacteroidota* cell from the Southern Temperate station stained with FLA-xylan. The third cell is *Planctomycetes* from the Southern Temperate station, which is stained with FLA-chondroitin. Cells were identified using the FISH probe PLA46 (magenta), which labels the riboplasm; the substrate staining is in the paryphoplasm.

Reintjes, G., Arnosti, C., Fuchs, B. et al. An alternative polysaccharide uptake mechanism of marine bacteria. *ISME J* 11, 1640–1650 (2017).
<https://doi.org/10.1038/ismej.2017.26>

Moreover, we routinely perform several tests to confirm the uptake is intracellular.

- Testing the staining on fixed cells (dead cells). No staining occurs with dead cells.

- *Testing whether the signal can be washed from cells using MQ, warm 1X PBS buffer, or artificial seawater. The FLAPS signal is still present even after washing.*
- *We have tested if fixed labelled cells will lose their signal when incubated with enzymes, which can degrade the specific FLAPS. We found no reduction of signal with enzyme incubations.*
- *We have performed FISH on FLAPS-stained cells, and the FLAPS signal remains even after the extensive FISH procedure. We see a reduction of the signal after the CARD-FISH procedure, primarily due to the permeabilisation and ethanol wash steps.*

All of these findings demonstrate that the signal is intracellular and specific.

Note that we have also shown that staining does not occur in cells which do not have the genetic potential (or mutants that have the potential removed) to take up the individual substrates, as demonstrated by Hehemann et al. (2019) Single cell fluorescence imaging of glycan uptake by intestinal bacteria. ISME J, 13(7), 1883-1889. <https://doi.org/10.1038/s41396-019-0406-z>

2) I think the 2 major strengths of this manuscript are the seawater analysis (Figures 3 & 4A) and the tips on distinguishing background signal from selfish cells (discussed at length in the Supplementary Information), which appear to be very important for the accuracy of this workflow while also being quite complex and sample-specific. I suggest the authors consider moving some of the background signal discussion into the main paper and figures, and also provide more microscopic images of fluorescent background (non-cellular) particles to compare to FLAPS-labeled cells, with the goal of helping other researchers distinguish them. The data in Figure S3B-C clearly show a difference between selfish cells and background signal, but the background signal is very high and sample-specific. Would researchers need to perform this comparison for every sample? The images in Figure S5 are helpful, but the manuscript could be improved with zoomed-in images of individual FLAPS-labeled cells compared to non-cellular fluorescent background particles. The resolution in the Figure 2 Step 6 image is closer to what I think you need here. The background signal discussion makes me think that the workflow presented in this manuscript is not "simple" as stated in the Figure 2 caption, but rather something that must be done with care and precision (see lines 307-308). That is fine, but I think the manuscript could emphasize this further and make the helpful tips in the Supp Info more prominent, as they are non-trivial. This is particularly important in light of my previous major comment about distinguishing FLAPS simply bound to organic debris (or even cells) rather than actually taken up into cellular periplasm.

We moved the entire comparison of tetra-labeled FISH and CARD-FISH and the section on non-specific staining into the primary methods section, as suggested. Line 433 - 491, Line 506 - 514 and Line 559 - 598

Minor comments:

1) General: Please clarify throughout the manuscript that the selfish uptake mechanism described herein only applies to Gram-negative bacteria (i.e., cells that contain an outer membrane and periplasmic space). Also please comment on whether Gram positive or Gram resistant bacteria can undergo selfish uptake (perhaps through a different mechanism?) and

whether the methods in this manuscript can detect that. If not, please discuss the limitations in investigating mixed community samples.

The current concept of a “selfish” uptake requires a cell organisation with an outer membrane that is lacking in gram-positive bacteria. To date, we have only once shown staining by FLAPS in Gram-positive bacteria (Bifidobacterium and Lactobacillus). This staining was very different from the halo-like staining in gram-negative bacteria. It appeared more like the whole cell was stained. However, the staining level was 50-fold less compared to gram-negative bacteria. Further investigations are still running to decipher this finding, and we do not have any more knowledge to date.

King ML, Xing X, Reintjes G, Klassen L, Low KE, Alexander TW, Waldner M, Patel TR, Wade Abbott D. In vitro and ex vivo metabolism of chemically diverse fructans by bovine rumen Bifidobacterium and Lactobacillus species. Anim Microbiome. 2024 Sep 9;6(1):50. doi: 10.1186/s42523-024-00328-1. PMID: 39252059; PMCID: PMC11382395.

In any case, we note that most bacteria in marine systems are gram-negative.

2) General (figures): Please give each panel its own letter to reduce confusion. For example, split Figure 3G into G (left image) and H (right image).

We have changed the Figure and given each panel its own letter.

Fig. 3 Polysaccharide utilization pattern in surface water from Helgoland in September over the course of a 72 h incubation. **(A)** Laminarin and xylan uptake stained a high proportion of cells already from the beginning of the incubation while **(B)** extracellular hydrolysis was comparatively rapid for xylan and chondroitin sulfate. **(C)** Microbial cell counts developed differently by substrate after the initial 24 h incubation. The **(D)** incubation-dependent changes of the initial community composition over 72 h show a FLA-chondroitin-dependent increase in *Flavicella* reads, but otherwise the community composition remained unchanged. Taxonomic identification with FISH showed **(E)** a large contribution of *Bacteroidetes* (CF319a) to selfish laminarin uptake while **(F)** a more diverse array of organisms including *Gammaproteobacteria* (Gam42a) and *Verrucomicrobia* (EUB338-III) took up xylan. **(G+H3)** Super-resolution structured illumination images showing different polysaccharide accumulation pattern after 24 hours incubation in FLA-laminarin stained cells (green), counterstained with DAPI (blue) and taxonomic correlation with FISH probes (red).

3) Lines 45-46: Please add that fungi also contribute substantially to degradation of plant polysaccharides in the environment (i.e., cellulose degradation), with appropriate references.

We have revised our statement: Polysaccharide degradation, transformation, and remineralization is mainly performed by bacteria, which are abundant in the environment ¹⁴ and

*in the digestive tracts of animals*¹⁵. *In terrestrial ecosystems, fungi also contribute substantially to the degradation of plant polysaccharides, such as cellulose*^{16,17, 18}.
Reference 16-18 are

Petr Baldrian, Vendula Valášková, Degradation of cellulose by basidiomycetous fungi, FEMS Microbiology Reviews, Volume 32, Issue 3, May 2008, Pages 501–521, <https://doi.org/10.1111/j.1574-6976.2008.00106.x>

Mäkelä, M. R., Donofrio, N., & de Vries, R. P. (2014). Plant biomass degradation by fungi. Fungal Genetics and Biology, 72, 2-9. <https://doi.org/10.1016/j.fgb.2014.08.010>

Berlemont, R. Distribution and diversity of enzymes for polysaccharide degradation in fungi. Sci Rep 7, 222 (2017). <https://doi.org/10.1038/s41598-017-00258-w>

4) Figure 1D: Please explain this image in more detail. Based on reading the text, I assume these are individual cells with DAPI in blue and FLAPS in green surrounding the cell. This was not immediately clear from the figure caption. (I didn't know if this was something inside an individual cell, outside an individual cell, etc.)

We have edited the figure legend: (D) Microscopic visualization of selfish FLAPS uptake and accumulation in the cells. The cell DNA is shown by DAPI staining in blue; the FLAPS is shown in green. Scale bar = 1µm

5) Figure 1E: Please clarify in the figure caption that this represents the external hydrolysis mechanism, not selfish uptake. Polysaccharides are being degraded into smaller products outside the cells, in the filtrate.

We have edited the figure legend as follows: (E) Gel permeation chromatogram showing systematic changes in molecular weight of FLAPS with incubation time (0 – 144 h). This figure shows the chromatograms used to measure the external (outside of the cell) hydrolysis of FLAPS. (Adapted from Arnosti, 2003³⁰).

6) Lines 91-93: Please specifically explain which parts of this manuscript are new compared to the previous Giljan et al. 2022 paper.

We have amended our statement to the following:

“These data, specifically the analysis of selfish uptake, FISH and extracellular hydrolysis rates, were initially presented in Giljan et al. (2022²⁹)” Line 134 - 135.

7) Line 93: Please clarify which data in this study are on human gut microorganisms. Is it the flow cytometry data in Figure 4B-C? If so, please clarify in the text and figure caption.

The data are taken from human gut microorganisms.

*“In addition, visualizing selfish uptake can reveal cellular variability in substrate processing as shown in the bacterial strain *Bacteroidota thetaiotaomicron*, a human gastrointestinal tract bacteria (Fig. 4A). When *B. thetaiotaomicron* is incubated with FLA-yeast mannan (Fig. 4A), individual cells in bacterial cultures can exhibit a variable extent of staining. In some instances, all cells in a pure culture take up FLAPS to a similar extent, while in other instances, some cells are more strongly stained than others.” Line 257 -264*

We have also amended our Figure 4 legend to “**Fig. 4** Metabolic phenotyping and flow cytometry of human gut microorganisms cultures and complex communities”.

8) Line 169: Please reference Figure 3A.

We have now referenced the figure.

9) Line 185-187 & Figure 3G: I'm not sure I see an enlarged periplasmic space in either of these images. Please clarify. Both appear to show polar localization. Also, you need a way to prove that the FLAPS are indeed located in the periplasmic space (see my major comment about this).

The images show an enlarged periplasmic space. In gram-negative bacteria, we observe a halo-like staining in the cells when FLAPS is taken into the periplasm – as shown in image A below. It's a very even distribution within the cell. However, in some bacteria, we see a polar localisation or an enlarged periplasm (image B). In these cells, we observe a substrate signal along the outer edge of the cell, but there is a clear area showing no FISH (ribosome signal) or DAPI (DNA) signal.

Taken from - Reintjes, G., Arnosti, C., Fuchs, B. et al. An alternative polysaccharide uptake mechanism of marine bacteria. *ISME J* 11, 1640–1650 (2017).
<https://doi.org/10.1038/ismej.2017.26>

We added a further image of a Z-stacking of cells (below) showing an enlarged periplasm to highlight that it is through the cells.

“Supplementary Figure S7. SR-SIM Z-stack of cells incubated with chondroitin sulphate. Images (a-p) show horizontal slices of the cells at 0.2 μ m intervals. Chondroitin (green) is in the paryphoplasm; FISH signal (red, Pla46) is in the riboplasm, and DNA is stained by DAPI (blue). Scale bar = 1 μ m.”

Reintjes, G., Arnosti, C., Fuchs, B. et al. An alternative polysaccharide uptake mechanism of marine bacteria. *ISME J* 11, 1640–1650 (2017). <https://doi.org/10.1038/ismej.2017.26>

10) Figure 3C: It appears that cells started dying in the control and laminarin samples, but continued to grow in the chondroitin samples. Does this imply that selfish uptake does not support population growth as well as external hydrolysis (compare Figure 3C to 3A & 3B)? If so, please comment on the relative importance of these 2 mechanisms in this environment.

Total cell counts decrease only in the control and laminarin incubations. However, these are environmental samples; we do not know the extent to which viruses or grazers were present and active in our incubations. It is possible that the selfishly active organisms increase in abundance and are then subject to viral lysis. It is also possible that because the selfish organism uses the substrate “for themselves,” other organisms die due to insufficient nutrient availability or secondary metabolites from the selfish cells. However, we have no evidence to support this hypothesis, as our focus was the degradation of the FLAPS.

The results of Figures 3A and 3B show that both mechanisms are active, including selfish uptake of laminarin after 48-72 hours, as cell counts decrease. This finding indicates that selfish organisms are not declining in abundance; instead, other community members are decreasing in number. In the chondroitin sulfate incubation, it is possible that the extracellular degradation causes the production of public goods, leading to growth of the community. However, the only data we have is the change in cell abundance, and the fact that low molecular weight hydrolysis products were present in the seawater. We do not know whether viruses or grazers were equally active in the different incubation.

For further information about the relative balance between selfish uptake and external hydrolysis in oceanic bacterial communities and the factors that may affect this balance, please see Brown et al. (2023) Pulsed inputs of high molecular weight organic matter shift the mechanisms of substrate utilization in marine bacterial communities. *Env Microb*, doi: 10.1111/1462-2920.16580, as well as Arnosti et al. (2021) The biogeochemistry of marine polysaccharides: Sources, inventories, and bacterial drivers of the carbohydrate cycle. *Annual Review of Marine Science*, vol. 13. doi: 10.1146/annurev-marine-032020-012810. This is currently an active area of research for us.

11) Figure 3 caption: Please quantify the terms "increase," "unchanged," etc.

To clarify the point that with the exception of the 72 h chondroitin sample, relative community composition was similar among incubations, we changed the wording to “The (D) incubation-dependent changes of the initial community composition over 72 h show an increase in relative contribution of *Flavicella* reads at the 72h timepoint in the chondroitin incubations, but otherwise the relative community composition remained similar among incubation and control samples and changed little over 72 h.”

12) Lines 192-194 & Figure 4A: Please describe specifically how the images show heterogeneous vs. homogeneous uptake patterns, and exactly what these terms mean in this context. The images are difficult to interpret.

Changed the text in the caption to the following:

(A) Visualization of cell-to-cell variability in extent of staining of strains of *Bacteroidota thetaiotaomicron* incubated with fluorescently labeled yeast mannan. Note how the FLAPS staining is variable among individual cells in the left panel, while the FLAPS staining is similar among all cells in the right panel. The cell DNA is shown by DAPI staining in blue, the FLAPS is shown in green and cell membranes are shown by Nile red staining in red. Scale bar = 5 μ m.

We also changed the labels for Fig 4A to read “Variable FLAPS uptake among cells” and “Similar FLAPS uptake among cells” to make the major points clear.

Fig. 4 Metabolic phenotyping and flow cytometry of human gut microorganisms cultures and complex communities. **(A)** Visualization of cell-to-cell variability in extent of staining of strains of *Bacteroides thetaiotaomicron* incubated with fluorescently labeled yeast mannan. Note how the FLAPS staining is variable among individual cells in the left panel, while the FLAPS staining is similar among all cells in the right panel. The cell DNA is shown by DAPI staining in blue, the FLAPS is shown in green and cell membranes are shown by Nile red staining in red. Scale bar = 5 μm **(B)** Quantification of FLAPS uptake into *Bacteroides thetaiotaomicron* by flow cytometry. Shown in the first two panels are negative controls: *Bacteroides thetaiotaomicron* grown on unlabeled monosaccharide and unlabeled polysaccharide. The subsequent three panels show the change in fluorescence of the cells with incubation in FLAPS for 5 min, 1 day and 3 days. Data revisualized from Klassen et al., 2021. The red dotted line represents the fluorescence threshold of the control community. **(C)** Schematic representation fluorescence-activated cell sorting of FLAPS positive cells in combination with sequencing.

Furthermore, we rewrote the paragraph containing those lines to emphasize the point that the extent of FLAPS uptake can vary among strains of bacteria, and can be visualized microscopically.

“When *B. thetaiotaomicron* is incubated with FLA-yeast mannan (Fig. 4A), individual cells in bacterial cultures can exhibit a variable extent of staining. In some instances, all cells in a pure culture take up FLAPS to a similar extent, while in other instances, some cells are more strongly stained than others. Identifying and monitoring possible differences in staining among cells is essential in applications where cells are used for bioprocesses (e.g., production of fuels such as ethanol, butanol, fatty acid derivatives or natural products), as it affects biosynthesis performance, specifically enzyme activity or expression level³⁵.” (Line 260 - 268)

13) Figure 4B: Please explain the significance of the red dotted lines.

*The red dotted line is the minimal threshold line, we have added this to the figure legend.
“The red dotted line represents the fluorescence threshold of the control community.”*

14) Figure 4C: This is a conceptual model. Can you perform this analysis on your seawater sample and show real data to demonstrate the utility of this method (combining flow cytometry with bacterial community profiling)?

Figure 4C is a conceptual model based on the similar analysis performed for the first time in 2012 in marine systems and again in 2022 in both marine and rumen samples. As the investigations use different substrates and samples from different sources we did not include the results but opted for a conceptual model.

We have highlighted these studies in the section Line 201 Flow cytometry: tracking specific organisms.

Martinez-Garcia, M., Brazel, D. M., Swan, B. K., Arnosti, C., Chain, P. S. G., Reitenga, K. G., Xie, G., Poulton, N. J., Gomez, M. L., Masland, D. E. D., Thompson, B., Bellows, W. K., Ziervogel, K., Lo, C.-C., Ahmed, S., Gleasner, C. D., Detter, C. J., & Stepanauskas, R. (2012). Capturing Single Cell Genomes of Active Polysaccharide Degradors: An Unexpected Contribution of Verrucomicrobiota . PLOS ONE, 7(4), e35314. <https://doi.org/10.1371/journal.pone.0035314>

Giljan, G., Arnosti, C., Kirstein, I. V., Amann, R., & Fuchs, B. M. (2022). Strong seasonal differences of bacterial polysaccharide utilization in the North Sea over an annual cycle. Environmental Microbiology, 24(5), 2333-2347. <https://doi.org/https://doi.org/10.1111/1462-2920.15997>

Klassen, L., Reintjes, G., Li, M., Jin, L., Amundsen, C., Xing, X., Dridi, L., Castagner, B., Alexander, T. W., & Abbott, D. W. (2022). Fluorescence activated cell sorting and fermentation analysis to study rumen microbiome responses to administered live microbials and yeast cell wall derived prebiotics [Original Research]. Front Microbiol, 13, 1020250. <https://doi.org/10.3389/fmicb.2022.1020250>

15) Line 220-221 & Figure 3G: Please specify the time difference between the 2 panels, either here or in the figure caption.

The images are from incubations after 24hr, we have added this to the figure legend.

16) Lines 226-241: These conclusions are very interesting and generally well supported by your seawater data (although see my first major comment).

Thank you.

17) Lines 242-262: I don't think this text is needed, as it does not describe conclusions from this particular study. The authors could move it to the Introduction if desired.

As suggested, we moved this text into the Introduction.

Reviewer #3 (Comments for the Author):

General comment - The authors have not presented data that shows "selfish" uptake, which infers nutrient acquisition by denying others access to the substrate. To accurately show a selfish mechanism the authors would need to have competition assays where two bacterial strains that can use a substrate do not get equal access to substrate. What they have done is develop a method to identify bacteria that can use uptake labeled polysaccharides and be able

to visualize them amongst a complex community. It is still a useful tool, but the FLAPS data as presented does not identify selfish bacteria, just those that can import a tagged substrate. Instead of trying to sell a story about identifying selfish bacteria, the authors should reframe the manuscript as a methods guide to using the FLAPS and highlight the various applications, but also add context about all of the extra instrumentation required to use FLAPS effectively (e.g. FISH, microscopy, FACS, etc.).

Cuskin et al. (2015) introduced the concept of 'selfish' uptake using pure cultures of bacteria, in which they indeed demonstrated that one (gut) bacterium obtained all of the substrate while not providing hydrolysis products to a second gut bacterium that was dependent upon hydrolysis products.

We have cited Cuskin et al. (2015) extensively because we have found that a similar uptake mechanism can be found among members of natural microbial communities in a wide range of locations in the ocean. As the reviewer might not be aware, it is currently not possible to isolate in pure culture the vast majority of bacteria in the ocean.

Rappé, M. S., & Giovannoni, S. J. (2003). "The Uncultured Microbial Majority." Annual Review of Microbiology, 57(1), 369-394. DOI: 10.1146/annurev.micro.57.030502.090759

Moreover, a major goal of our work is to reveal the mechanisms by which organic matter is cycled in the ocean, a process involving multiple members of microbial communities interacting in various ways. So trying to isolate specific bacteria and carry out the experiments the reviewer wants to see would in any case likely not be successful in a marine context, and moreover would not lead to answers to the questions that we are pursuing about carbon cycling in the ocean.

We use the term 'selfish' bacteria to denote bacteria that transport large fragments of polysaccharides into the periplasmic space – in other words, as a means of denoting the mechanism of degradation of polysaccharides. At least some of these bacteria are very likely 'selfish' in the sense that the reviewer views the term, since we cannot detect any hydrolysis products of these polysaccharides in seawater, despite the fact that they are taken up by the selfish bacteria ((Reintjes et. al., 2017)). Some polysaccharides, however, are hydrolyzed externally by a community and simultaneously taken up selfishly by some of the bacteria ((Reintjes et. al., 2019)). Simultaneous selfish uptake and external hydrolysis in seawater is quite common. We cannot, in fact, determine which community members in a seawater incubation are carrying out external hydrolysis – determining this is currently beyond the capabilities of the scientific community as a whole: measuring enzyme activities in a natural sample, rather than genetic potential to produce an enzyme, or production of specific proteins, is a challenge that has received far too little attention in the past decades. So we can identify the selfish bacteria, and can speculate about the identities of external hydrolyzers – this is the current state of the art.

We thus disagree with the reviewer with respect to which the term originated by Cuskin et al can be used in analysis of natural samples. We contend, however, that our writing is sufficiently clear that the reader can determine the context in which we are using the term 'selfish'.

We agree that we have a methods manuscript. A very detailed account of the method to produce and use FLAPS to measure external hydrolysis in environmental samples has long been published (Arnosti 2003), and this method has been used in more than 70 manuscripts since the publication of Arnosti (1996; A new method for measuring polysaccharide hydrolysis

rates in marine environments. *Organic Geochem.* 25:105-115.) We therefore focused the current manuscript specifically on identification of selfish bacteria – in our sense, not the sense of the reviewer – and use of various add-on analyses (FISH, flow cytometry) that can provide further information about the bacteria that use this uptake mechanism. Given the questions of all three reviewers, however, we have now moved the information about production of FLAPS into the main text, and have expanded it.

Associated edits to the manuscript:

Abstract: “this methods paper, we present a detailed guide to identifying selfish bacteria, including techniques for rapidly visualizing selfish uptake in complex bacterial communities, detecting selfish organisms, and distinguishing their activity from that of other community members” Line 26 – 29

Introduction: “Selfish’ bacteria ²¹ bind polysaccharides and partially hydrolyze them to oligosaccharides, which are transported into the periplasm and then undergo further degradation. This minimizes the release of mono-, di-, and trisaccharides into the surrounding environment, ensuring a return on their enzymatic investment. Line 69 – 72)

Results “Note that the concentrations of FLAPS added to seawater samples (3.5 μmol monomer-equivalent per liter of seawater) is aimed at producing detectable signals during gel permeation chromatography/fluorescence detection – in other words, to obtain a rate of extracellular hydrolysis – and represents a fairly low addition of organic carbon (typically on the order of 20 $\mu\text{mol C L}^{-1}$ seawater), especially considering typical organic carbon concentrations in the surface ocean and nearshore environments. Thus, the FLAPS are not intended as a growth substrate, although an increase in cellular abundance is sometimes observed, as in the current incubation for chondroitin and xylan. By the same measure, the fact that the incubations are carried out in unfiltered seawater means that grazers and viruses are present in the same incubations as the bacteria. A decrease in cell numbers during incubation, as observed in the laminarin and unamended incubations, can be due to the activities of grazers and/or viruses naturally present in seawater. Line 188 - 199

Methods: Section entitled: *Synthesis and characterization of fluorescently labeled polysaccharides (FLAPS) and Measurements of external extracellular hydrolysis.* Line 342 -390

p. 3 l. 121 - Why are the limnic and gut microbiome data not available in this paper? If the authors are mentioning them, they should be included. The authors have a sizable Supplemental Data document, so what is the rationale for mentioning, but then not sharing, those data? Nearly all of the unpublished data in Table S1 are for *B. thetaiotaomicrom*, are the authors expecting to publish these results later, and if so, why mention them now?

We have now added references to limnic studies in the introduction (Cackiovic 2025, Keith & Arnosti 2001; Ziervogel et al. 2014; Balmonte et al. 2016, 2019; Bullock et al. 2017). We have updated the table as many datasets have indeed now been published. Some of the unpublished references are in the process of being published, and they are in the list as unpublished to highlight the breadth of FLAPS available. The focus of the current manuscript is a methods paper – a how-to guide with information that cannot be published in the typical manuscript where the focus is on the results and interpretation, not on helping others apply a specific method.

p. 4 l. 124-125 - What is the rationale for the substrate choice (laminarin, xylan, and chondroitin)? The environmental sample is surface sea water, so why have one substrate that is more often considered terrestrial-plant derived? Are marine plants rich in xylan? If you wanted to use marine-plant specific substrates, wouldn't alginate or agarose be better choices?

The substrates we use represent a selection of polysaccharides that are available in marine environments, and that also fulfill the criteria listed in the SI (now in the main text):

“In principle, any soluble polysaccharide (or phytoplankton extract or phytoplankton-derived DOC; Arnosti, 2003; Arnosti et al., 2005; Murray et al., 2007) can be labeled using this procedure. The primary limitations on polysaccharide choice are the requirements for solubility in an aqueous solution and stability (and solubility) at pH > 9.5.”

Many interesting polysaccharides are either insoluble, or too gelatinous to label and use as FLAPS. As it happens, however, xylan is found in marine systems (also as part of marine plants as well as algae), and xylanase activity is periodically high in different locations, including in Helgoland, as shown in our example. We agree that alginate and agarose would be desirable choices, but both coagulate easily and are difficult to label, which restricts our ability to use them as FLAPS. We also note that sample analysis (for measurement of external hydrolysis) is very time consuming, so for many investigations, we use only a select suite of substrates due to limitations of time and resources.

p.5 l. 136 - Why is cell growth so poor? The authors say cell counts increased, but there is not even a doubling of cell number within 24 hours and is barely 3 cell doublings in 72 hours. Are FLAPS even viable substrates for growth? How much label is attached to a polysaccharide and how does that impact degradability?

We do not aim to cause cell growth with FLAPS addition. The FLAPS are added at a concentration just sufficient to generate a fluorescence signal in the gel permeation chromatography/fluorescence detection system. We deliberately try to minimize the addition of organic carbon added to incubations in order to minimize disturbance of the original microbial community. Cell growth is not the aim of this procedure – detecting external (extracellular) enzyme activities and identifying selfish cells is the main point. We have added text to the manuscript to make this important point more clearly. The FLAPS (when added at high concentrations) do indeed cause cell growth, but that is not our objective. We note also that natural bacterial communities in marine waters typically do not double rapidly unless they are in the presence of high concentrations of organic matter (as they are during a spring phytoplankton bloom, for example.) Moreover, rapid growth of natural microbial communities (as during phytoplankton blooms) also typically leads to rapid declines, due grazing activity and viruses.

Figure 3G - Why is the FLA-laminarin signal just at the poles of the bacteria? If the probe was periplasmic, shouldn't it be a ring as is seen in Figure 1D?

As noted in the Conclusions section, selfish uptake has been best studied in members of the Bacteroidotas, but we observe selfish uptake in organisms such as members of the Verrucomicrobiota that frequently show this type of staining pattern. The Verrucomicrobiota have a distinct periplasm, which is considered a cellular compartment located between the inner cytoplasmic membrane and the outer membrane. It has the unique feature of being enlarged in some members of the phylum in a mono-bipolar fashion.

Boedeker, C., Schüler, M., Reintjes, G. et al. Determining the bacterial cell biology of Planctomycetes. *Nat Commun* 8, 14853 (2017). <https://doi.org/10.1038/ncomms14853>

We see similar enlargement in the FLAPS staining patterns with cell we identified as belonging to the Phylum. See also image below which shows a substrate signal along the outer edge of the cell, but there is a clear area showing no FISH (ribosome signal) and no DAPI (DNA) signal.

B:

Taken from - Reintjes, G., Arnosti, C., Fuchs, B. et al. An alternative polysaccharide uptake mechanism of marine bacteria. *ISME J* 11, 1640–1650 (2017). <https://doi.org/10.1038/ismej.2017.26>

Figure 4. There is no mention at all of Nile Red staining in the text. The authors need to explain why it was done and what the results mean.

In Line 255-256 we state the use of Nile Red staining to visualize the cell membranes. “The use of a membrane stain can show the co-localization of the polysaccharide-associated green fluoresceinamine signal with the red membrane stain Nile Red in a fluorescent intensity line grating²⁷, further demonstrating polysaccharide uptake into the periplasmic space.”

Nile red staining is also mentioned in the methods section under super-resolution microscopy.

Supplementary Methods - As written there is not enough information for another researcher to generate FLAPS. This is just a rewording of the short description on p.8 l.280-285. Furthermore, there is a mention of unpublished results in the description to generate FLAPS. This also occurs in several other places in the manuscript. There is no reason to not include the missing information so the authors should include it.

We did not initially include extensive information about FLAPS synthesis in the main part of the manuscript since a manuscript describing the method in great detail (cited in the ms) was published more than 20 years ago:

Arnosti, C. (2003) Fluorescent derivatization of polysaccharides and carbohydrate-containing biopolymers for measurement of enzyme activities in complex media. *J. Chromatog. B* 793: 181-191.

Since that time, more than 70 manuscripts have been published that use FLAPS to measure external (extracellular) enzymatic hydrolysis of polysaccharides, of polysaccharides from dissolved organic matter excreted by phytoplankton, and of polysaccharide-containing phytoplankton extracts in seawater, freshwater, and marine sediments. We therefore did not consider this information to be especially novel. Given the remarks from all of the reviewers

asking for further information about FLAPS, however, we have moved the supplemental information about FLAPS production into the main text. Moreover, we have changed 'unpublished results' to 'personal communication', since we have no plans to publish separately individual observations that we have made in the course of producing FLAPS over the last several decades. We include them here, however, since some of these observations will help others to avoid spending time with fluorophores or protocols that are unlikely to work out (e.g., using fluorophores that have an ester linkage in the side chain of the fluorophore connected to a polysaccharide results in a linkage that is unstable in aqueous solution.)

Re: Spectrum01602-24R1 (Using phenotyping to visualize and identify selfish bacteria: a methods guide)

Dear Dr. Greta Reintjes:

Thank you for the privilege of reviewing your work. Below you will find my comments, instructions from the Spectrum editorial office, and the reviewer comments.

Dear authors

the reviewer raised significant criticismplease be aware of them and try to review your MS in line with theirs suggestions

Revision Guidelines

Sincerely,
Sandi Orlic
Editor
Microbiology Spectrum

Reviewer #2 (Comments for the Author):

Thank you for your detailed responses to my previous comments.

My major concerns about the original manuscript have been satisfactorily addressed by the revisions. Specifically, moving the discussion of selfish uptake from the Conclusions to the Introduction was very helpful in addressing reader questions upfront.

The text clearly addresses the distinction between selfish uptake vs. binding to the outside of the cell, and the references to previous studies (including the z-stack data) where this has been clearly demonstrated are provided right away.

Please address these additional minor comments:

1) Please add the discussion of Gram-positive vs Gram-negative bacteria to the manuscript text. The information you provided in the Response to Comments (Reviewer 2, Minor comment 1) is helpful, but I didn't see it added to the text. I think this information is important for readers to understand the prevalence of selfish uptake.

2) Please add the information provided in the Response to Comments (Reviewer 2, Minor comment 9; Reviewer 3, Figure 3G comment) to the discussion of Figure 3G in lines 242-251. Please include a reference to Reintjes et al. 2017 here, emphasizing that this polar localization has been documented previously and shown to indicate an enlarged periplasmic space, with a clear explanation of why this is the case.

Subject: Spectrum01602-24 Decision Letter

Re: Spectrum01602-24 (Using phenotyping to visualise and identify selfish bacteria: a hunting guide)

Dear Editor,

We thank the reviewers for their further constructive comments and have addressed them in detail below. *Our responses are in italics and blue font.* We have added a mark up document to show the two changes in the text.

Reviewer #2 (Comments for the Author):

Thank you for your detailed responses to my previous comments.

My major concerns about the original manuscript have been satisfactorily addressed by the revisions. Specifically, moving the discussion of selfish uptake from the Conclusions to the Introduction was very helpful in addressing reader questions upfront. The text clearly addresses the distinction between selfish uptake vs. binding to the outside of the cell, and the references to previous studies (including the z-stack data) where this has been clearly demonstrated are provided right away.

Please address these additional minor comments:

1) Please add the discussion of Gram-positive vs Gram-negative bacteria to the manuscript text. The information you provided in the Response to Comments (Reviewer 2, Minor comment 1) is helpful, but I didn't see it added to the text. I think this information is important for readers to understand the prevalence of selfish uptake.

Line 118: Additionally, the current concept of a "selfish" uptake requires a cell organization with an outer membrane and is therefore restricted to gram-negative bacteria.

2) Please add the information provided in the Response to Comments (Reviewer 2, Minor comment 9; Reviewer 3, Figure 3G comment) to the discussion of Figure 3G in lines 242-251. Please include a reference to Reintjes et al. 2017 here, emphasizing that this polar localization has been documented previously and shown to indicate an enlarged periplasmic space, with a clear explanation of why this is the case.

Line 241: To date cell staining has shown two distinct patterns. Halo-like staining within the entire periplasmic space shows an even signal seen in Gammaproteobacteria and Bacteroidota. Polar staining with one or both ends of the cell showing a clear increased signal, sometimes associated with an enlargement of the periplasmic space, observed in Verrucomicrobiota and Plantomycetes²⁹.

Re: Spectrum01602-24R2 (Using phenotyping to visualize and identify selfish bacteria: a methods guide)

Dear Dr. Greta Reintjes:

Your manuscript has been accepted, and I am forwarding it to the ASM production staff for publication. Your paper will first be checked to make sure all elements meet the technical requirements. ASM staff will contact you if anything needs to be revised before copyediting and production can begin. Otherwise, you will be notified when your proofs are ready to be viewed.

Sincerely,
Sandi Orlic
Editor
Microbiology Spectrum

Reviewer #2 (Comments for the Author):

Thank you for addressing the additional comments.